# ZNF185 prevents stress fiber formation through the inhibition of RhoA in endothelial cells

Soichiro Suzuki[1], Fumiaki Ando [1 ✉], Sae Kitagawa[1], Yu Hara[1], Tamami Fujiki[1], Shintaro Mandai [1], Koichiro Susa[1], Takayasu Mori[1], Eisei Sohara[1], Tatemitsu Rai[1] & Shinichi Uchida[1]

Signaling through cAMP/protein kinase A (PKA) promotes endothelial barrier function to prevent plasma leakage induced by inflammatory mediators. The discovery of PKA substrates in endothelial cells increases our understanding of the molecular mechanisms involved in vessel maturation. In this study, we evaluate a cAMP inducer, forskolin, and a phospho-PKA substrate antibody to identify ZNF185 as a PKA substrate. ZNF185 interacts with PKA and colocalizes with F-actin in endothelial cells. Both ZNF185 and F-actin accumulate in the plasma membrane region in response to forskolin to stabilize the cortical actin structure. By contrast, ZNF185 knockdown disrupts actin filaments and promotes stress fiber formation without inflammatory mediators. Constitutive activation of RhoA is induced by ZNF185 knockdown, which results in forskolin-resistant endothelial barrier dysfunction. Knockout of mouse *Zfp185* which is an orthologous gene of human *ZNF185* increases vascular leakage in response to inflammatory stimuli in vivo. Thrombin protease is used as a positive control to assemble stress fibers via RhoA activation. Unexpectedly, ZNF185 is cleaved by thrombin, resulting in an N-terminal actin-targeting domain and a C-terminal PKA-interacting domain. Irreversible dysfunction of ZNF185 protein potentially causes RhoA-dependent stress fiber formation by thrombin.

[1] Department of Nephrology, Tokyo Medical and Dental University (TMDU), Tokyo 113-8510, Japan. ✉email: fandkidc@tmd.ac.jp

Endothelial cells line the tunica intima of blood vessels and organize the endothelial barrier to prevent plasma leakage from blood vessels into the interstitium. This barrier function is disrupted in patients with inflammatory diseases including sepsis and anaphylaxis[1,2]. Several inflammatory mediators, such as vascular endothelial growth factor, histamine, and thrombin, are responsible for the impairment of intercellular junction integrity, which results in vascular hyperpermeability[3].

Vascular endothelial (VE)-cadherin is an adhesion molecule, which is important for maintaining endothelial barrier function. It is specifically expressed in endothelial cells. The cytoplasmic tail of VE-cadherin forms a complex with catenins, which are attached to the actin cytoskeleton. Its extracellular domain homophilically binds to neighboring endothelial cells in a calcium-dependent manner to preserve the integrity of the endothelial junction[4]. Cell–cell junctions are often described as linear structures along the boundaries of endothelial cells; however, extracellular stimuli can reorganize junctional complexes. In response to inflammatory stimuli, contractile actin bundles, known as stress fibers, are generated from the submembranous actin cortex. VE-cadherin then redistributes into a zig-zag pattern with intercellular gaps, which results in endothelial hyperpermeability[5,6].

Cyclic adenosine monophosphate (cAMP) signaling is a common pathway that protects the endothelial barrier. In endothelial cell lines, cAMP-dependent kinase (protein kinase A, PKA) is activated by cAMP and counteracts endothelial dysfunction caused by inflammatory mediators[7–9]. RhoA is a downstream effector of PKA and enhances endothelial contractility. Inhibition of RhoA activity via PKA facilitates endothelial relaxation and tightens endothelial cell–cell junctions[10,11]. By contrast, PKA inhibition increases RhoA activity and induces F-actin assembly and the formation of stress fibers, thus promoting intercellular gap formation in the endothelium[12]. In vivo analyses revealed that decreased PKA activity in tumor endothelial cells results in vascular hyperpermeability and extravasation of plasma proteins[13]. Although PKA is a major effector of vascular permeability, the molecular mechanisms underlying the PKA signaling pathway in endothelial cells remain undefined.

PKA is a heterotetrameric holoenzyme consisting of two regulatory (PKA R) and two catalytic (PKA C) subunits. Concerning the PKA R subunit, four isoforms (RIα, RIβ, RIIα, and RIIβ) have been identified. Direct binding of cAMP to each isoform induces a conformational change and alters the subcellular distribution of the PKA C subunits. This triggers the phosphorylation of PKA substrates that contain the sequence, RRXS/T[14,15]. The RRXS/T motif is not randomly phosphorylated by PKA because the target specificity of PKA is spatiotemporally determined via its interacting proteins, such as A-kinase anchoring protein (AKAP). More than 70 functionally distinct AKAP proteins serve as subcellular address tags to create specific compartmentalized PKA signaling at each subcellular location[16]. Therefore, the identification of PKA substrates or PKA-interacting proteins is important to elucidate the molecular mechanisms that regulate the local PKA signaling network.

In the present study, we discovered that zinc-finger protein (ZNF) 185 exhibits characteristics of not only a PKA substrate but also a PKA-interacting protein. ZNF185 is considered a ZNF because it contains C-terminal *Lin-ll*, *Isl-1*, and *Mec-3* (LIM)-type zinc-finger domains, and its structure is maintained using zinc ions. The LIM domain acts as an interface that mediates protein–protein interactions[17]. ZNF185 also contains an N-terminal-located actin-targeting domain. It colocalizes with F-actin and cytoskeletal-related components including lamellipodia and filopodia[18]. ZNF185 is involved in the regulation of cell differentiation and cancer cell proliferation. Downregulation of ZNF185 expression is associated with the tumorigenesis, development, and metastasis of prostate and lung cancers[19,20]. By contrast, ZNF185 expression is upregulated in colon cancer and its expression is correlated with liver metastasis[21]. However, its physiological relevance in endothelial cells has not been determined. In the present study, we demonstrate that ZNF185 is required for the organization of endothelial cell–cell junctions and the suppression of vascular hyperpermeability by regulating the cAMP/PKA/RhoA signaling pathway.

## Results

**Identification of ZNF185 as a PKA substrate in HUVECs**. We used a phospho-PKA (pPKA) substrate antibody as a screening tool to identify PKA substrates in human umbilical vein endothelial cells (HUVECs). First, the cAMP/PKA activator, forskolin, was administered to HUVECs to induce the phosphorylation of endogenous PKA substrates. The phosphorylation status of PKA substrates was evaluated via Western blot analysis using a pPKA substrate antibody, which recognizes phosphoproteins containing the RRXpS/T motif. As shown in Fig. 1a, forskolin phosphorylated multiple PKA substrates, which are indicated by black arrows. Immunoprecipitation with a pPKA substrate antibody was done to isolate these pPKA substrates. Immunoprecipitated pPKA substrates, indicated by asterisks, were detected by silver staining (Fig. 1b). They were subsequently excised from the gels and subjected to mass spectrometry (Supplementary Fig. 1). Among the list of identified proteins, we focused on ZNF185 as a putative pPKA substrate because ZNF185 contains three distinct PKA phosphorylation motifs, RRXS (Supplementary Fig. 2a).

Stable HUVEC cell lines overexpressing Myc-ZNF185-HA were established to determine the phosphorylation status and intracellular localization of ZNF185. An anti-ZNF185 antibody detected endogenous ZNF185 indicated by a black arrowhead as well as overexpressed ZNF185 indicated by arrows (Fig. 1c). The band intensity of endogenous ZNF185 was reduced by ZNF185 knockdown (Supplementary Fig. 3). We confirmed that overexpressed ZNF185 was phosphorylated by forskolin (Fig. 1c), similar to that of endogenous ZNF185 (Fig. 1b, Supplementary Fig. 1). Forskolin rapidly phosphorylated ZNF185 within 5 min (Fig. 1d). Immunofluorescence staining revealed that pZNF185 (indicated by white arrows), which was detected using both Myc and pPKA substrate antibodies, emerged in the membrane region of HUVECs after the administration of forskolin (Fig. 1e). VE-cadherin was used as a membrane marker. These results indicate that forskolin activated PKA in HUVECs, which subsequently phosphorylated ZNF185 localized at the cell membrane.

**ZNF185–PKA interaction promotes ZNF185 phosphorylation**. As shown in Fig. 2a, forskolin-induced phosphorylation of ZNF185 indicated by arrowheads was attenuated by the PKA inhibitor, Myr-PKI, confirming that ZNF185 was phosphorylated by PKA. The intracellular localization, activity, and substrate specificity of PKA are generally regulated using interacting proteins. Colocalization of PKA with its downstream cellular targets, mediated by anchoring proteins, determines the substrate specificity of PKA[15,22]. This suggests that the PKA substrate, ZNF185, is close to PKA. Therefore, we performed a coimmunoprecipitation assay to establish an interaction between ZNF185 and PKA. ZNF185 interacted with PKA RIIα, and RIIβ out of the four PKA R isoforms (Fig. 2b).

To identify the interaction region between ZNF185 and PKA RII subunits, we generated four ZNF185 fragments (Fig. 2c). A coimmunoprecipitation assay revealed that PKA RIIα interacted with the C-terminal fragment (aa 501–689) of ZNF185 (Fig. 2d). In line with previous reports[18], actin interacted with the N-terminal fragment of ZNF185 (aa 1–200). Based on this result, we generated a ZNF185 deletion mutant (Δaa 534–605) (Supplementary Fig. 2b). Unlike the full-length ZNF185, neither

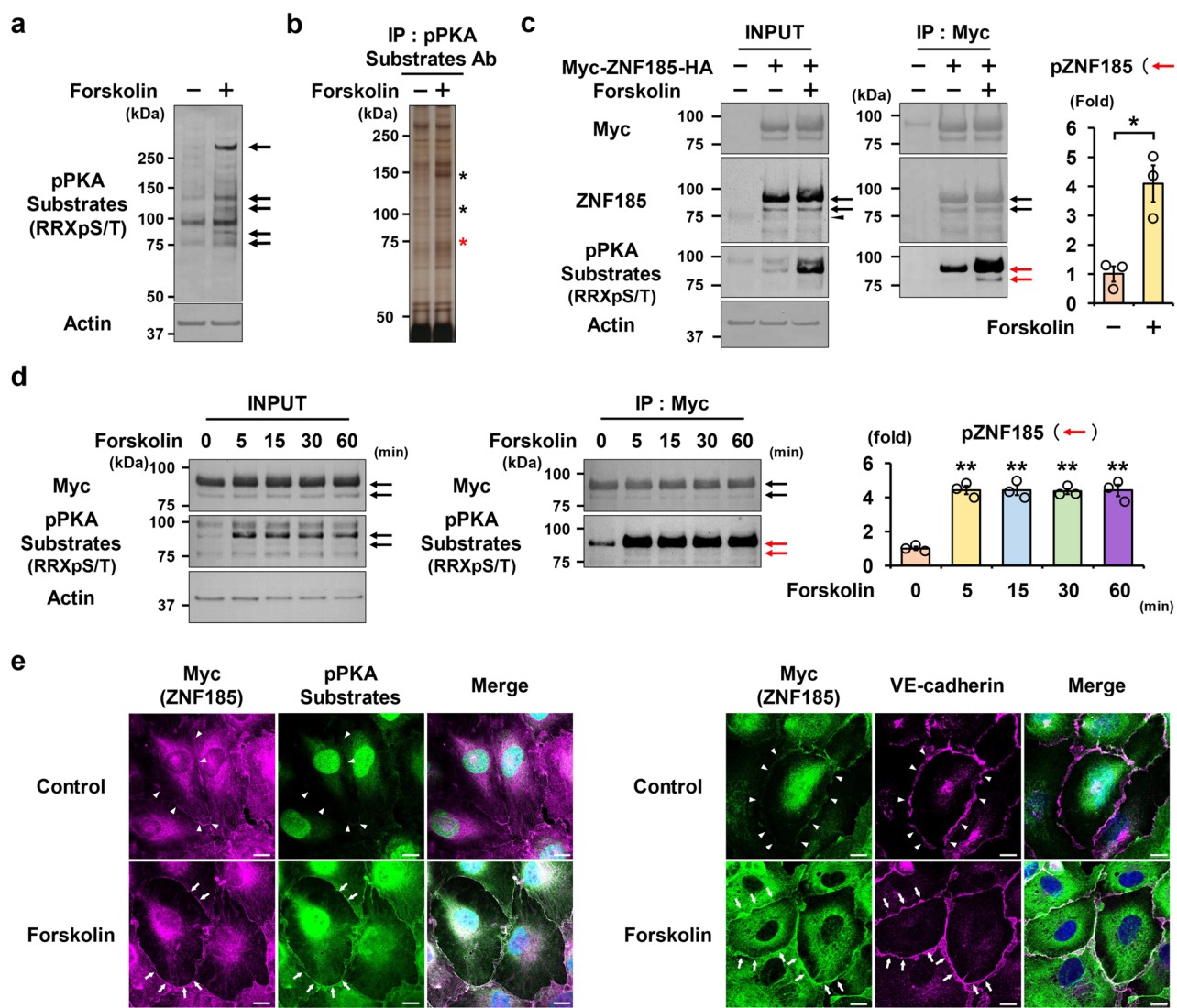

**Fig. 1 Identification of ZNF185 as a PKA substrate in HUVECs. a** Several PKA substrates are phosphorylated using forskolin in HUVECs. Forskolin (10 µM) was added to HUVECs for 1 h. pPKA substrates are indicated by arrows. Representative blots are shown ($n = 3$). **b** Representative silver-staining of the immunoprecipitated PKA substrates in HUVECs. Forskolin (10 µM) was added to the HUVECs for 1 h. pPKA substrates indicated by asterisks were immunoprecipitated by a pPKA substrate antibody ($n = 3$). The band indicated by a red asterisk was ZNF185 (Supplementary Fig. 1). **c** Overexpressed ZNF185 was phosphorylated by forskolin in HUVECs. Stable cell lines of HUVECs overexpressing Myc-ZNF185-HA were generated. One hour after the administration of forskolin (10 µM), anti-Myc beads were used for immunoprecipitation. Densitometric analysis of pZNF185. The black arrows indicate overexpressed ZNF185 and the arrowhead indicates endogenous ZNF185. The red arrows indicate phosphorylated ZNF185. *$p < 0.05$ ($n = 3$). **d** Overexpressed ZNF185 was phosphorylated by forskolin within 5 min. Time course of ZNF185 phosphorylation after the administration of forskolin (10 µM) is shown. Anti-Myc beads were used for immunoprecipitation. Densitometric analysis of pZNF185. The arrows indicate overexpressed ZNF185. **$p < 0.01$ ($n = 3$). **e** Overexpressed pZNF185 is localized to the membrane region. Immunofluorescence staining of pPKA (green) and Myc (magenta) in HUVECs overexpressing Myc-ZNF185-HA is shown. pZNF185 accumulates at the membrane region after the administration of forskolin (10 µM) for 1 h. Arrowheads indicate the cell boundaries. Arrows indicate pZNF185 localized to the membrane region. VE-cadherin was used to stain the cell membrane. Representative images are shown ($n = 3$). Scale bars, 10 µm. Data are presented as the mean ± standard error.

PKA RIIα nor PKA RIIβ interacted with the ZNF185 deletion mutant (ZNF185-ΔPKA) (Fig. 2e, Supplementary Fig. 4). As a result of the dissociation between ZNF185 and PKA, the phosphorylation of ZNF185-ΔPKA following forskolin treatment was impaired (Fig. 2f). This indicates that both the PKA inhibitor (Fig. 2a) and ZNF185-ΔPKA (Fig. 2f) negatively influence ZNF185 phosphorylation. The ZNF185–PKA interaction is required for the phosphorylation of ZNF185 via PKA.

**Stress fiber assembly by ZNF185 knockdown**. In endothelial cell lines, actin structures are altered dynamically using cAMP/PKA

signaling. cAMP/PKA creates a cortical actin rim to enhance the stability of the cell–cell junction[23]. By contrast, PKA inhibitors promote intercellular gap formation[12]. Consistent with previous studies, ZNF185 interacted with actin in HUVECs despite the presence of forskolin (Figs. 2d, 3a). In addition, ZNF185 and F-actin were colocalized (Supplementary Fig. 5). In response to forskolin treatment, both ZNF185 and F-actin were translocated to the membrane region and comprised the cortical actin rim. These results suggest that ZNF185 is involved in actin cytoskeletal structures in HUVECs.

Next, we examined the role of ZNF185 in endothelial barrier integrity. ZNF185 knockdown induced actin rearrangement,

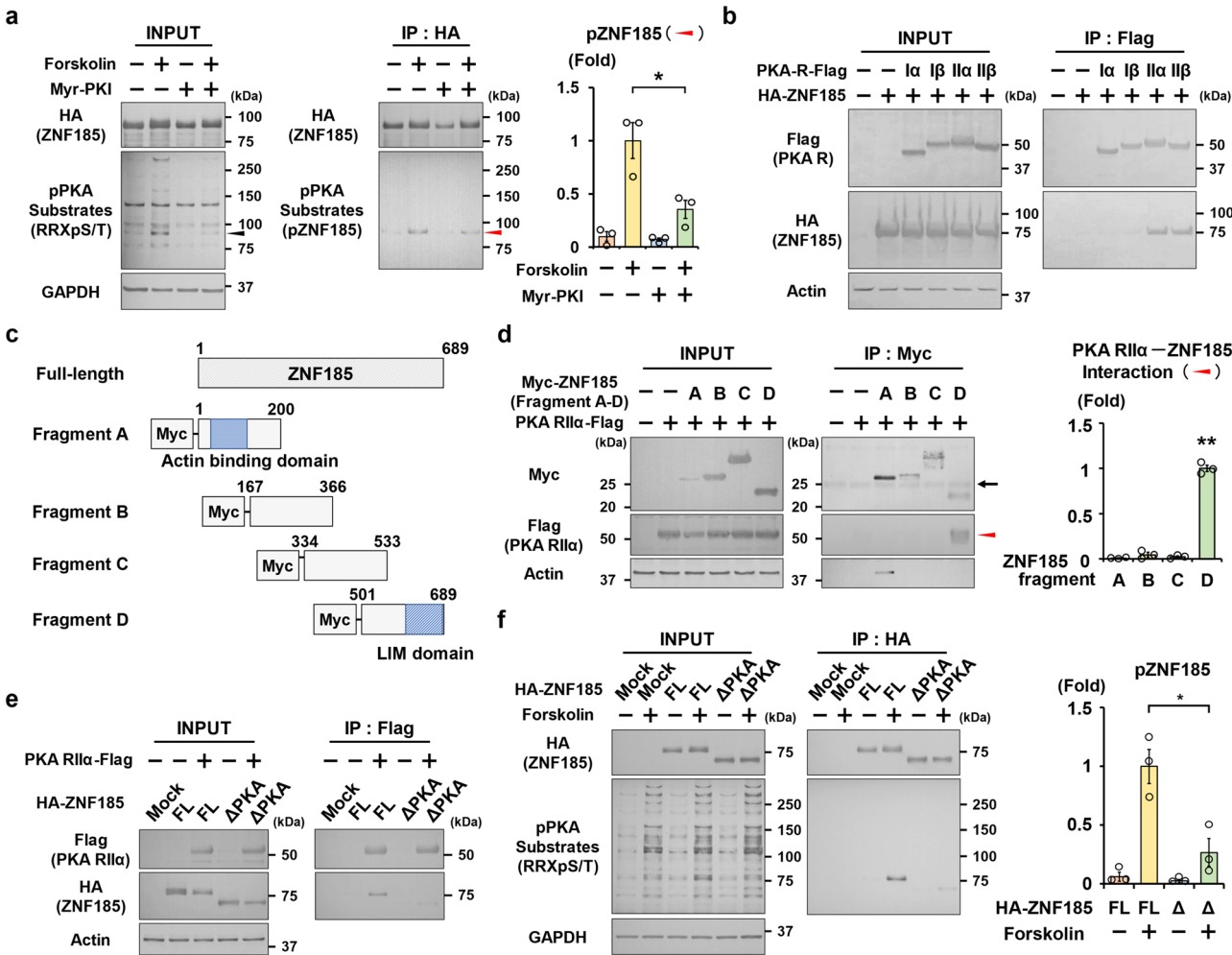

**Fig. 2 The ZNF185–PKA interaction is responsible for ZNF185 phosphorylation. a** Phosphorylation of ZNF185 is inhibited via Myr-PKI. HUVECs overexpressing Myc-ZNF185-HA were pretreated with Myr-PKI for 1 h before forskolin stimulation for 1 h. Anti-HA-beads were used for immunoprecipitation. The arrowheads indicate phosphorylated ZNF185. Densitometric analysis of pZNF185. Two-sided Student's t test, *$p < 0.05$ ($n = 3$). **b** ZNF185 interacts with PKA RIIα and PKA RIIβ. PKA regulatory subunits (RIα, RIβ, RIIα, and RIIβ)-Flag and HA-ZNF185 are overexpressed in HEK293T cells. Anti-Flag beads were used for coimmunoprecipitation. Representative blots are shown ($n = 3$). **c** Schematic representation of ZNF185 fragments. **d** PKA RIIα-Flag interacts with the C-terminal fragment of ZNF185. PKA RIIα-Flag and Myc-ZNF185 fragments are overexpressed in HEK293T cells. Anti-Myc beads were used to perform coimmunoprecipitation. Representative blots are shown. The black arrow indicates light chain. The red arrow indicates immunoprecipitated PKA RIIα. Densitometric analysis of PKA RIIα-Flag coimmunoprecipitated with Myc-ZNF185 fragments. **$p < 0.01$ ($n = 3$). **e** PKA RIIα does not interact with the deletion mutant of ZNF185 (Δaa 534–605). PKA RIIα-Flag and HA-ZNF185 are overexpressed in HEK293T cells. Anti-Flag beads were used for coimmunoprecipitation. Representative blots are shown ($n = 3$). **f** ZNF185-ΔPKA is not phosphorylated via forskolin. One hour after the administration of forskolin (10 μM), anti-HA beads were used for coimmunoprecipitation. Densitometric analysis of pZNF185. *$p < 0.05$ ($n = 3$). Data are presented as the mean ± standard error. FL full-length, Δ Δaa 534–605.

causing severely organized actin stress fibers in a confluent monolayer of HUVECs (Fig. 3b–d, Supplementary Fig. 6a, b). We further evaluated VE-cadherin dynamics at endothelial junctions. Immunofluorescence staining of VE-cadherin revealed that ZNF185 knockdown formed discontinuous zig-zag junctions indicated by white arrows with inter-endothelial gap junctions indicated by white arrowheads (Fig. 3b, Supplementary Fig. 6c). cAMP/PKA promotes the accumulation of VE-cadherin at cell boundaries, leading to an improvement in endothelial cell–cell adhesion[24]. Forskolin treatment prevented stress fiber formation in control HUVECs transduced with scrambled shRNA; however, actin stress fibers did not fully recover in ZNF185 knockdown HUVECs. These results indicate that ZNF185 is necessary for stable endothelial barrier function.

**Enhanced endothelial permeability by ZNF185 knockdown.** Endothelial barrier dysfunction increases the leakage of solutes across endothelial cells. A quantitative Transwell permeability assay was done to measure FITC-dextran permeability through a tight monolayer of HUVECs. The permeability was decreased using forskolin treatment and increased using the PKA inhibitor, Myr-PKI (Fig. 4a), which is consistent with previous studies[25,26]. Thrombin was used as a positive control. ZNF185 knockdown exhibited similar effects as Myr-PKI and strongly increased permeability to the same extent as thrombin (Fig. 4b). Consistent with the immunofluorescent results shown in Fig. 3b, the barrier stabilizing effects of forskolin were attenuated in ZNF185 knockdown HUVECs compared with control HUVECs. Results indicate that ZNF185 as well as cAMP/PKA are important elements for the maintenance of endothelial permeability.

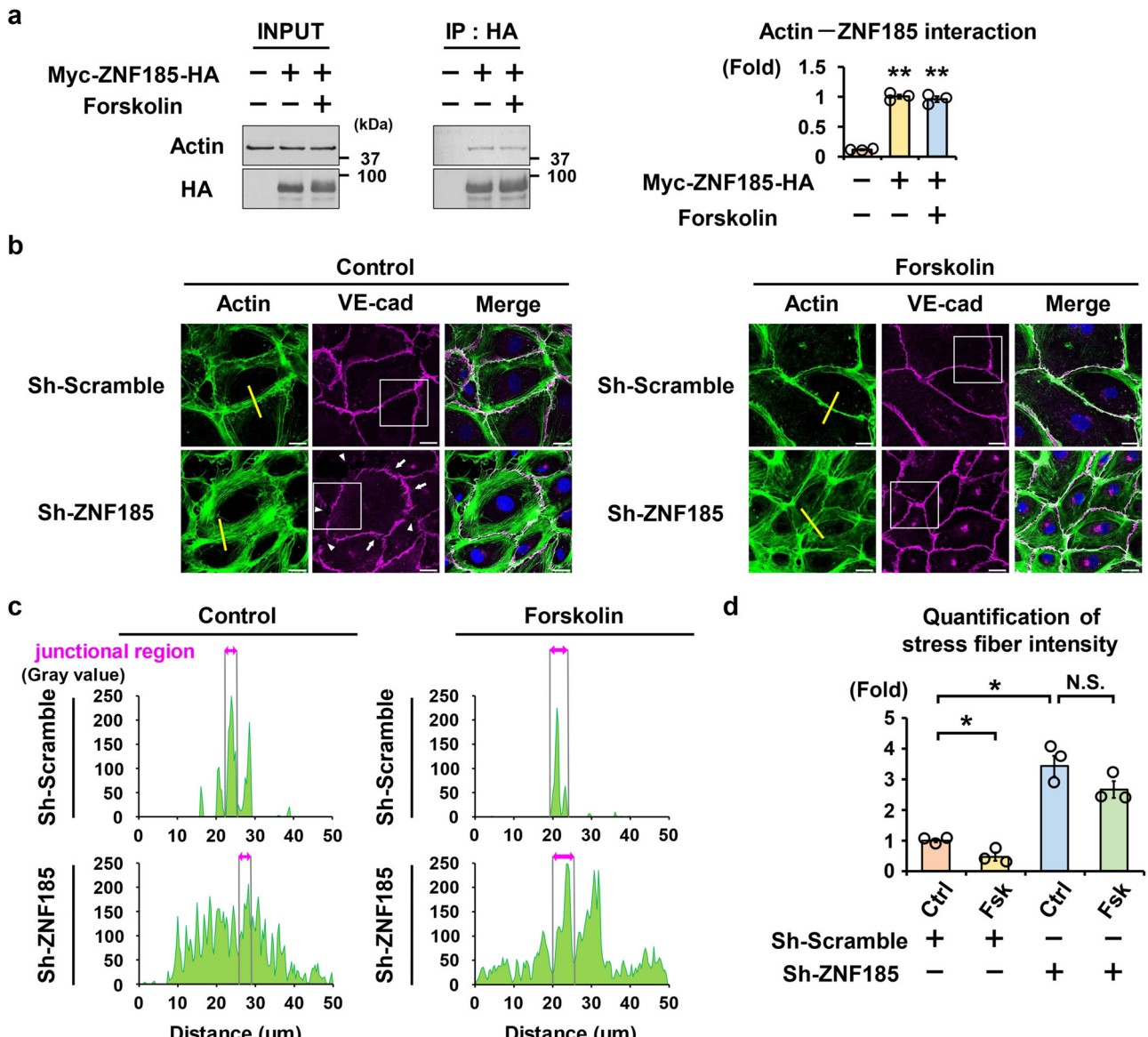

**Fig. 3 ZNF185 knockdown induces stress fiber formation. a**. ZNF185 interacts with actin. Myc-ZNF185-HA was overexpressed in HUVECs. One hour after forskolin (10 μM) treatment, HA-beads were used for coimmunoprecipitation. Representative blots are shown ($n = 3$). Densitometric analysis of immunoprecipitated endogenous actin. Two-sided Student's $t$ test, $**p < 0.01$ ($n = 3$). **b–d** ZNF185 knockdown induces stress fiber formation and discontinuous cell–cell junctions in a confluent HUVECs monolayer. **b** Immunofluorescence staining of VE-cadherin (magenta) and actin (green). Arrows indicate discontinuous zig-zag junctions. Arrowheads indicate intercellular gaps. Representative images are shown ($n = 3$, Supplementary Fig. 6). Scale bars, 25 μm. **c** Intensity of stress fibers at the yellow lines in Fig. 3b was quantified. **d** The bar graph shows total amount of stress fiber intensity in Fig. 3c. Two-sided Student's $t$ test, $*p < 0.05$ ($n = 3$). Data are presented as the mean ± standard error. Ctrl control, Fsk forskolin.

Next, we determined the effect of ZNF185 knockdown on RhoA activity because PKA/RhoA signaling stabilizes endothelial barrier function[10,25] and ZNF185 knockdown and Myr-PKI exhibited similar effects on endothelial permeability (Fig. 4a, b). ZNF185 knockdown significantly increased the GTP-bound active form of RhoA in HUVECs (Fig. 4c), indicating that ZNF185 suppressed basal RhoA activity like that of PKA. Inhibition of RhoA/Rho-associated protein kinase signaling by Y-27632 inhibited baseline stress fiber assembly (Fig. 4d, Supplementary Fig. 7a, b) and endothelial permeability (Fig. 4e) in sh-scrambled HUVECs, as previously reported[27,28]. Y-27632 successfully counteracted stress fiber formation via ZNF185 knockdown (Fig. 4d) and potently ameliorated forskolin-resistant hyperpermeability caused by ZNF185 knockdown (Fig. 4e). These results indicate that ZNF185 was required for

cAMP/PKA-induced RhoA inhibition and endothelial barrier stabilization.

RhoGDIα is a well-known mediator of PKA/RhoA signaling pathway. DerMardirossian et al. and Qiao et al. demonstrated that PKA phosphorylates RhoGDIα at S174[29,30], which increases the binding affinity between RhoGDIα and RhoA and then sequesters RhoA to suppress RhoA activity[31]. RhoA is conversely activated by *RhoGDIα* knockout, leading to elevation of basal endothelial permeability in vivo[32]. In addition, knockdown of RhoGDIα in endothelial cell lines disrupts adherens junctions and promotes stress fiber formation[32]. In line with previous reports, we confirmed that knockdown of RhoGDIα by sh-RhoGDIα activated RhoA (Fig. 5a) and induced stress fiber formation in HUVECs (Fig. 5b–d). PKA activation by forskolin did not improve stress fiber formation, indicating that RhoGDIα was an

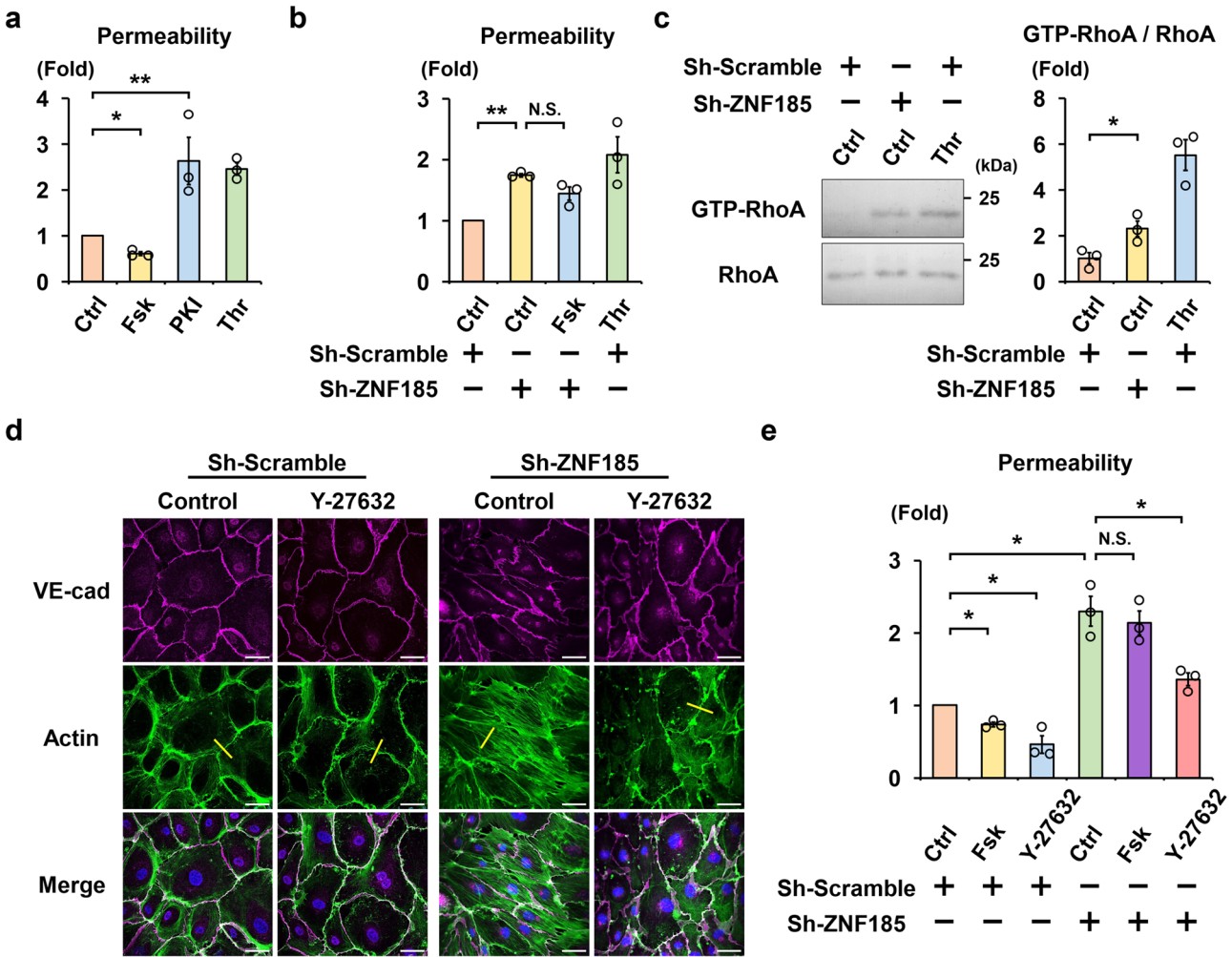

**Fig. 4 ZNF185 knockdown increases endothelial permeability via RhoA. a** Endothelial permeability changes in response to PKA activity in HUVECs. Permeability was determined by measuring the passage of FITC-dextran via confluent HUVECs monolayers. Thrombin was used as a positive control. Two-sided Student's $t$ test, $*p < 0.05$, $**p < 0.01$ ($n = 3$). **b** ZNF185 knockdown increases endothelial permeability in HUVECs. After knockdown of ZNF185 by Sh-ZNF185, 10 μM forskolin or 10 U/ml thrombin was administered to HUVECs for 1 h. Permeability was measured as in Fig. 4a. $*p < 0.05$ ($n = 3$). **c** RhoA is activated using ZNF185 knockdown. RhoA activity in HUVECs was measured using a rhotekin pull-down assay. Representative blots are shown. $*p < 0.05$ ($n = 3$). **d** Y-27632 improves ZNF185 knockdown-induced stress fiber formation in HUVECs. Y-27632 (10 μM) was added to HUVECs transduced with scrambled shRNA or ZNF185 shRNA for 1 h. Immunofluorescence staining of VE-cadherin (magenta) and actin (green). Representative images are shown ($n = 3$). Scale bars, 50 μm. Intensity of stress fibers at the yellow lines was quantified in Supplementary Fig. 7a. **e** Y-27632 attenuates endothelial hyperpermeability induced by ZNF185 knockdown. Y-27632 (10 μM) was added to HUVECs transduced with scrambled shRNA or ZNF185 shRNA for 1 h. Permeability was measured as in Fig. 4a. $*p < 0.05$ ($n = 3$). Data are presented as the mean ± standard error. Ctrl control, Fsk forskolin, PKI PKI 14-22 amide, myristoylated, Thr thrombin.

indispensable mediator of PKA/RhoA/actin stress fibers signaling pathway. We next examined the effects of ZNF185 knockdown on phosphorylation of RhoGDIα-S174 because ZNF185 interacted with PKA (Fig. 2b–e), and RhoGDIα-S174 is phosphorylated by PKA[29,30]. Although forskolin phosphorylated RhoGDIα-S174 in sh-scrambled HUVECs, its phosphorylation levels were consistently suppressed by ZNF185 knockdown regardless of the administration of forskolin (Fig. 5e). These results indicate that ZNF185 was essential for PKA-induced phosphorylation of RhoGDIα-S174, leading to inhibition of RhoA activity and stress fiber formation.

**Enhanced vascular permeability in *Zfp185* knockout mice.** Mouse *Zfp185* is an orthologous gene of human *ZNF185*[33]. Similar to ZNF185, Zfp185 interacted with PKA subunits (RIα, RIβ, RIIα, and RIIβ) as well as actin (Fig. 6a, b). We then generated *Zfp185* knockout mice using CRISPR/Cas9 genome editing

system (Fig. 6c–e, Supplementary Fig. 8). Evans blue dye was used to measure vascular permeability. The topical application of mustard oil to the right ear, which induces cutaneous inflammation, enhanced Evans blue dye leakage in *Zfp185* knockout mice (Fig. 6f, g). These results indicate that Zfp185 is involved in vascular permeability under inflammatory conditions.

**Cleavage of ZNF185 by thrombin.** Thrombin changes cytoskeletal structures through Rho-dependent and Rho-independent (calcium/calmodulin-mediated) pathways and serves as a positive control for stress fiber formation in HUVECs[34,35]. However, the precise molecular mechanisms concerning how thrombin activates RhoA are unknown. Thrombin is a protease and ZNF185 contains the thrombin-susceptible sequence, LTPRAGLR[36]. Thrombin promoted stress fiber formation in a dose-dependent manner (Fig. 7a) and 3 U/ml of thrombin was sufficient to cleave endogenous ZNF185 in HUVECs (Fig. 7b). In addition, thrombin

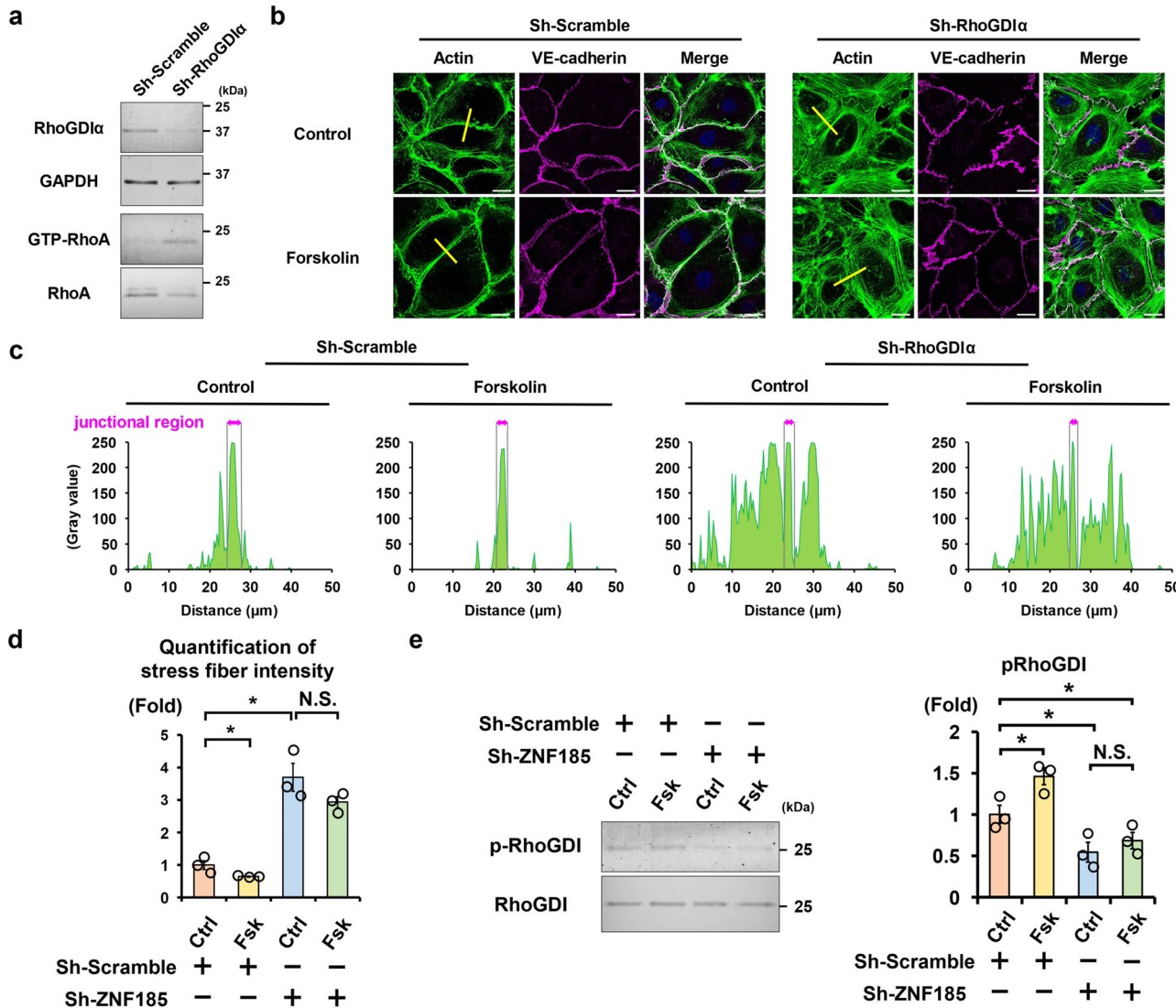

**Fig. 5 ZNF185 is essential for PKA-induced phosphorylation of RhoGDIα. a** RhoGDIα knockdown induces RhoA activation. RhoA activity in HUVECs was measured using a rhotekin pull-down assay. Representative images are shown ($n = 3$). **b–d**. RhoGDIα knockdown induces stress fiber formation in a confluent HUVECs monolayer. **b** Immunofluorescence staining of actin (green) and VE-cadherin (magenta). Representative images are shown ($n = 3$). Scale bars, 25 μm. **c** Intensity of stress fibers at the yellow lines in Fig. 5b was quantified. **d** The bar graph shows total amount of stress fiber intensity in Fig. 5c. Two-sided Student's $t$ test, *$p < 0.05$ ($n = 3$). **e** ZNF185 knockdown suppressed RhoGDIα phosphorylation. Western blots and densitometry analysis of phospho-RhoGDI. Representative blots are shown. Two-sided Student's $t$ test, *$p < 0.05$ ($n = 3$). Data are presented as the mean ± standard error. Ctrl control, Fsk forskolin.

efficiently cleaved overexpressed Myc-ZNF185-HA (Fig. 7c). Immunoprecipitated recombinant Myc-ZNF185-HA was also cleaved by thrombin (Fig. 7d). By contrast, overexpressed ZNF185, which lacks the LTPRAGLR sequence located in exons 11 and 12 (ZNF185-Δexon 11,12) (Fig. 7e), was not cleaved by thrombin (Fig. 7f). To investigate the role of ZNF185 cleavage in HUVECs, we generated stable cell lines overexpressing full-length ZNF185 or ZNF185-Δexon 11,12. Although thrombin induced stress fiber formation and the disruption of endothelial adherens junctions in HUVECs overexpressing full-length ZNF185, ZNF185-Δexon 11,12 rescued these disruptive effects of thrombin (Fig. 8a, Supplementary Fig. 9a, b).

ZNF185 was irreversibly divided into an N-terminal actin-targeting domain and a C-terminal PKA-interacting domain by thrombin. Importantly, both fragments lost their physiological function. We overexpressed N-terminal ZNF185 (aa 1–245) and C-terminal ZNF185 (aa 305–689) in HUVECs (Fig. 8b).

Consistent with the results of full-length ZNF185 (Fig. 1e), N-terminal ZNF185 (green) was translocated to the membrane region in response to forskolin (Fig. 8c). By contrast, C-terminal ZNF185 (magenta) was dispersed throughout the cytosol and did not respond to forskolin. These results indicate that cleaved C-terminal fragment of ZNF185 was mislocalized by thrombin.

Intracellular localization of N-terminal ZNF185 was not impaired. We next examined whether N-terminal ZNF185 lacking PKA-interacting domain was sufficient to inhibit stress fiber formation. Overexpression of N-terminal ZNF185 with ZNF185 knockdown failed to phosphorylate RhoGDIα-S174 (Fig. 8d). As a result, ZNF185 knockdown-induced stress fiber formation was not rescued by overexpressed N-terminal ZNF185 (Fig. 8e, Supplementary Fig. 9c, d). These results indicate that both N- and C-terminal ZNF185 were indispensable for maintaining physiological function of ZNF185. Thrombin-induced irreversible cleavage of ZNF185 was one of the causes of stress fiber formation in HUVECs.

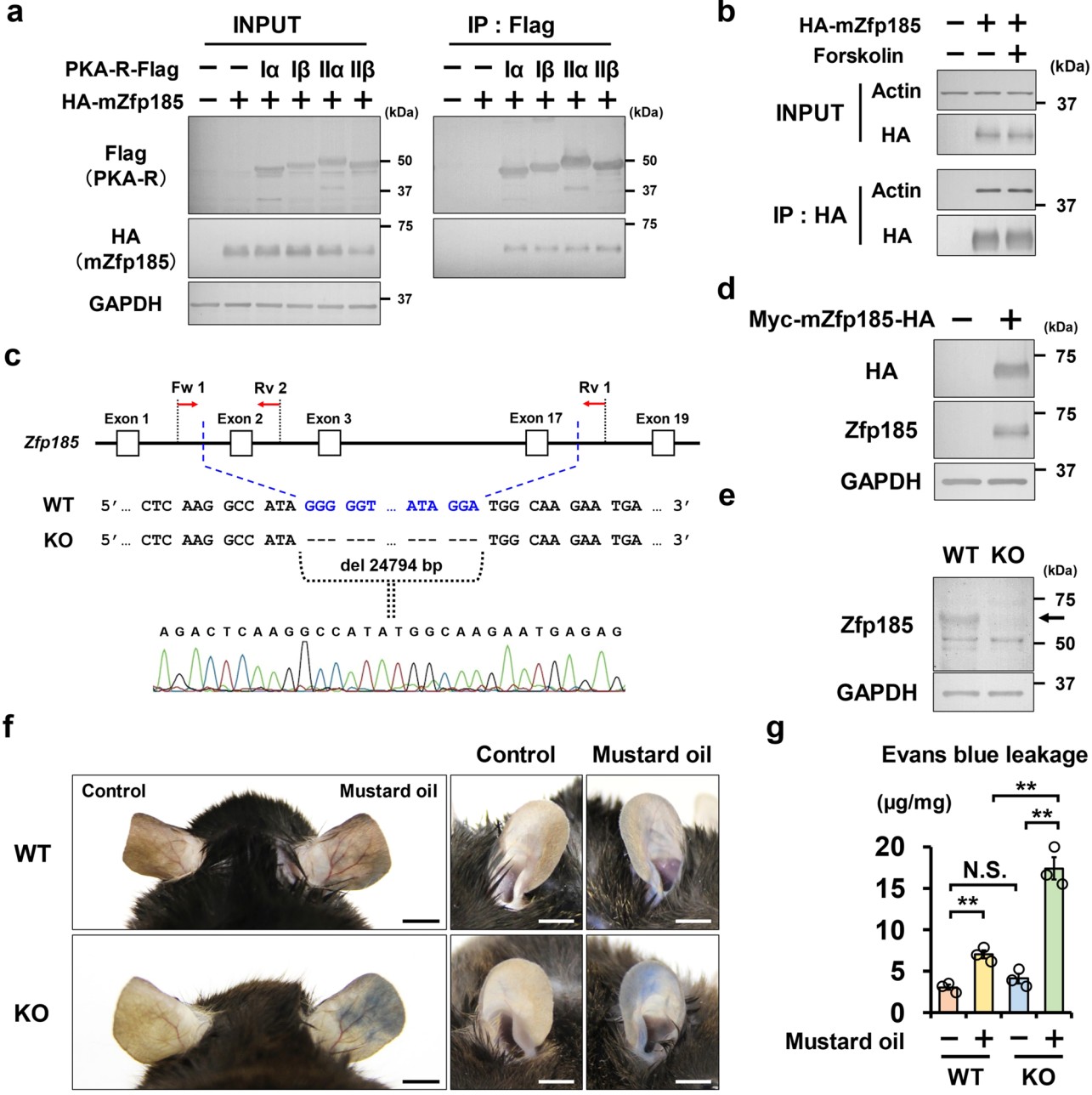

**Fig. 6 Vascular permeability is enhanced in *Zfp185* knockout mice. a**, **b** mZfp185 interacts with PKA regulatory subunits (RIα, RIβ, RIIα, and RIIβ) as well as actin. PKA regulatory subunits (RIα, RIβ, RIIα, and RIIβ)-Flag and HA-mZfp185 are overexpressed in HEK293T cells. Anti-Flag beads were used for coimmunoprecipitation. Representative blots are shown ($n = 3$). **c** Generation of *Zfp185* knockout mice by CRISPR/Cas9 genome editing technology. The target sequence for *Zfp185* gene editing is shown. Sanger sequencing confirms 24794 bp deletion. **d**, **e** mZfp185 is detected by anti-ZNF185 antibody. **d** Myc-mZfp185-HA were overexpressed in the HEK293T cells. Anti-ZNF185 antibody was used to detect overexpressed Zfp185. Representative blots are shown ($n = 3$). **e** Anti-ZNF185 antibody detects Zfp185 in the ear in WT mice ($n = 3$). **f** Representative images of Evans blue leakage. Inflammatory mustard oil increased Evans blue leakage in *Zfp185* knockout mice. Scale bars, 5 mm. **g** Quantification of Evans blue leakage in ears. Two-sided Student's *t* test, **$p < 0.01$ ($n = 3$). Data are presented as the mean ± standard error.

## Discussion

In the present study, we demonstrated that ZNF185 prevents stress fiber formation in endothelial cells by inhibiting RhoA (Fig. 9, Supplementary Fig. 10). We identified ZNF185 as a PKA substrate using forskolin and a pPKA substrate antibody (Fig. 1a, b, Supplementary Fig. 1). The development of methods to screen PKA substrates in endothelial cells has attracted attention because PKA is a therapeutic target in the promotion of vessel maturation[37]. Compared with previous chemical genetic approaches[38], immunoprecipitation with a pPKA substrate

antibody represented a rapid and efficient screening method to uncover pathogenic PKA substrates. ZNF185 was phosphorylated by PKA (Figs. 1c, 2a). By contrast, inhibition of the ZNF185–PKA interaction via ZNF185-ΔPKA strongly decreased ZNF185 phosphorylation (Fig. 2e, f). ZNF185 is the membrane-targeting PKA substrate in endothelial cells. Forskolin treatment phosphorylated ZNF185, which translocated it from the cytosol to the membrane region (Fig. 1e).

The C-terminal region of ZNF185 interacts with PKA (Fig. 2c, d), whereas the N-terminal region of ZNF185 proved to be an

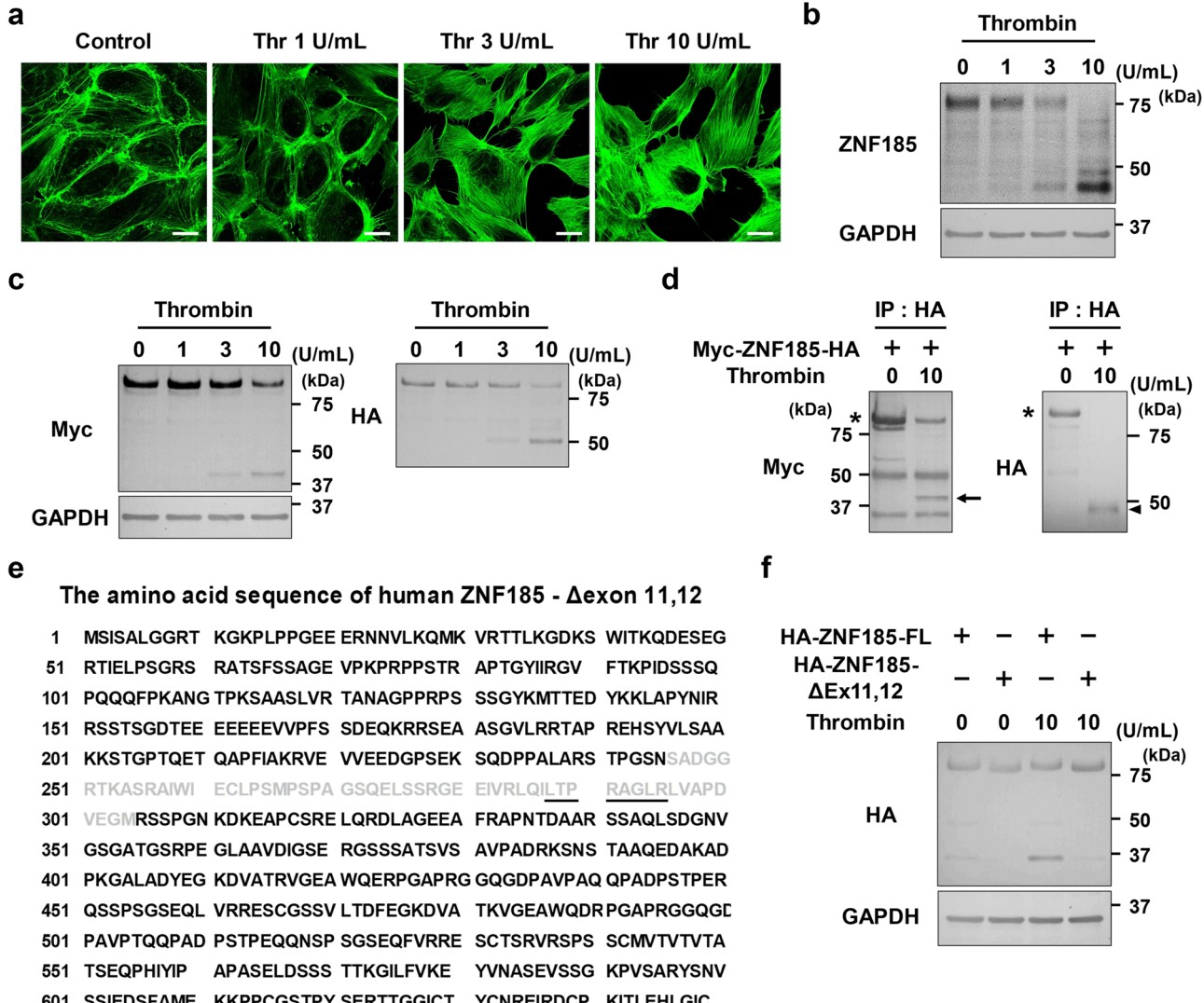

**Fig. 7 ZNF185 is cleaved via thrombin. a, b** Thrombin cleaves endothelial ZNF185 in HUVECs. **a** Thrombin promotes stress fiber formation in a dose-dependent manner. Immunofluorescence staining of actin (green) is shown ($n = 3$). Scale bars, 25 μm. **b** Thrombin was administered to HUVECs for 1 h. Representative blots are shown ($n = 3$). **c** Thrombin cleavage of overexpressed ZNF185 in HUVECs. Thrombin was administered to HUVECs overexpressing Myc-ZNF185-HA for 1 h. Representative blots are shown ($n = 3$). **d** Thrombin cleaves recombinant ZNF185. Anti-HA beads were used to immunoprecipitate overexpressed Myc-ZNF185-HA in HUVECs. After immunoprecipitation, 10 U/ml thrombin was administered to recombinant ZNF185 for 1 h. Representative blots are shown ($n = 3$). The arrow indicates the N-terminal region of ZNF185 cleaved by thrombin. The arrowhead indicates the C-terminal region of ZNF185 cleaved by thrombin. Representative blots are shown ($n = 3$). **e** The amino acid sequence of human ZNF185-Δexon 11,12 is shown. Exon 11 and 12 are shaded. The thrombin recognition sequence is underlined. **f** Thrombin does not cleave ZNF185-Δexon 11,12 which lacks a thrombin recognition sequence. Representative blots are shown ($n = 3$). Thr thrombin, FL full-length, Δ ZNF185-Δexon 11,12.

actin-targeting domain (Fig. 2c, d; 3a)[18]. In HUVECs, ZNF185 and F-actin were colocalized and both contributed to stabilizing cortical actin structures induced by forskolin treatment (Fig. 3b–d, Supplementary Fig. 5). Cytoskeletal linker proteins that interact with F-actin have been thoroughly studied to clarify the mechanisms of endothelial barrier function. Of these, plectin has characteristics that are very similar to ZNF185. Plectin interacts with actin[39] and plectin deficiency upregulates stress fiber formation, leading to vascular leakage[40]. In the present study, plectin was identified as a ZNF185 interacting protein and interacts with the N-terminal region of ZNF185 (Supplementary Fig. 11a–d). Overexpressed plectin and ZNF185 were well colocalized at the membrane region (Supplementary Fig. 11e). The functional role of plectin was quite similar to that of ZNF185. Knockdown of plectin also induced stress fiber formation in HUVECs (Supplementary Fig. 12a, b). Furthermore,

phosphorylation of PKA substrates at the membrane region indicated by white arrows were markedly suppressed by knockdown of plectin or ZNF185 (indicated by white arrowheads) (Supplementary Fig. 12c). ZNF185, plectin, actin, and PKA may form a large complex at the membrane region of endothelial cells, which regulates the balance between stress fiber formation and endothelial barrier protection, depending on the cAMP/PKA signaling pathway.

RhoA inhibition by cAMP/PKA signaling protects against endothelial barrier dysfunction[10,41]. By contrast, both ZNF185 knockdown and thrombin activate RhoA (Fig. 4c) and induce endothelial hyperpermeability (Fig. 4a, b). Not only Y-27632 but also forskolin ameliorates thrombin-induced leaky junctions in endothelial cells. However, in these previous studies, low-dose thrombin <1 U/ml was used[10,42,43]. We demonstrated that high-dose thrombin above 3 U/ml cleaved endogenous ZNF185,

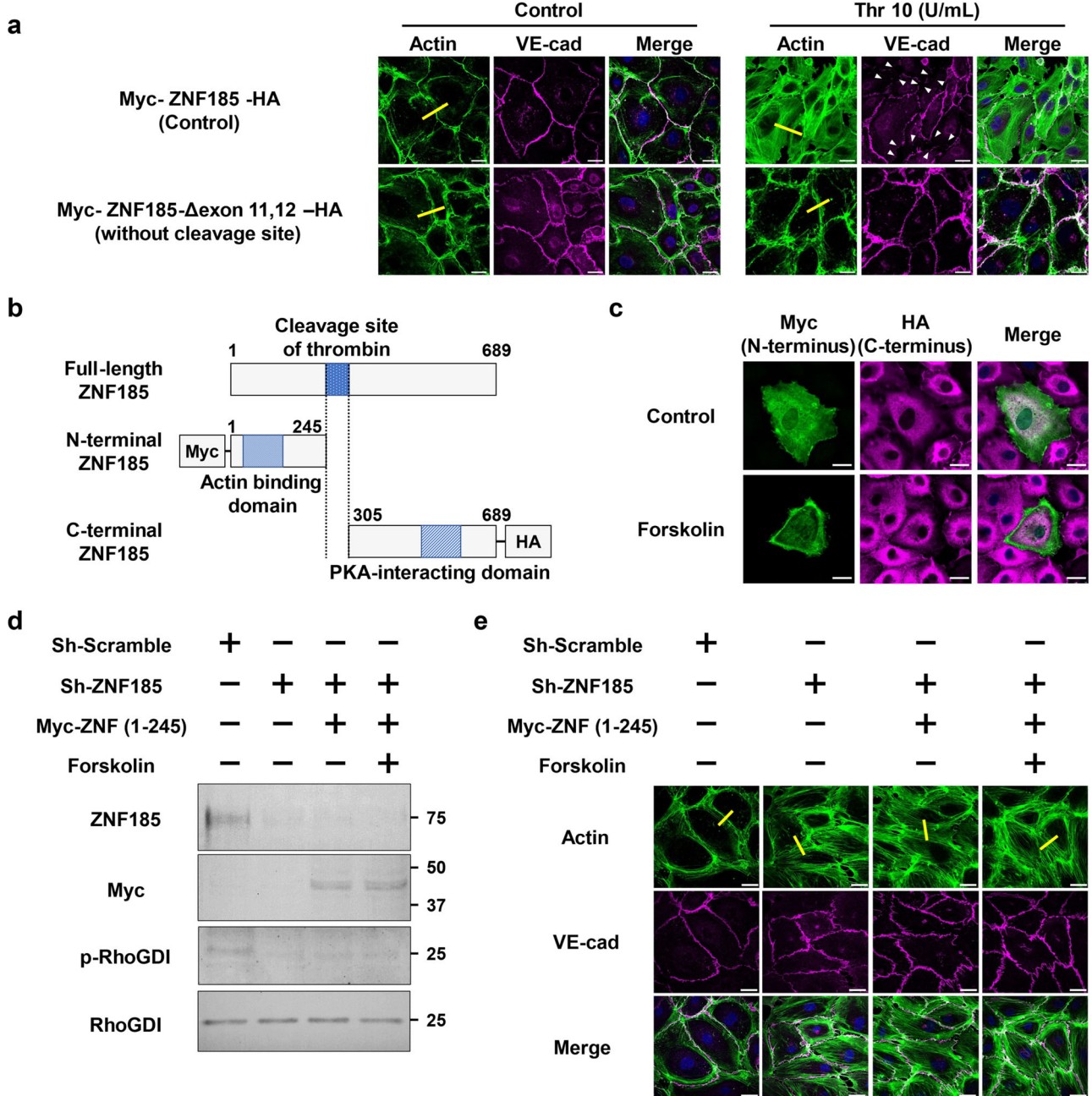

**Fig. 8 N-terminal ZNF185 lacking PKA-interacting domain is insufficient to inhibit stress fiber formation. a** Thrombin-induced stress fiber formation and disruption of adherens junction are inhibited by ZNF185-Δexon 11,12. Thrombin (10 U/mL) was added to HUVECs overexpressing Myc-ZNF185-HA or Myc-ZNF185-Δexon 11,12-HA for 1 h. Immunofluorescence staining of actin (green) and VE-cadherin (magenta). Arrowheads indicate intercellular gaps. Representative images are shown (*n* = 3). Scale bars, 25 μm. **b** Schematic representation of N- and C-terminal of ZNF185. **c** Myc-N-terminal ZNF185 (green) is translocated to the membrane region after the treatment of forskolin for 1 h. C-terminal ZNF185-HA (magenta) is dispersed throughout the cytosol (*n* = 3). Scale bars, 25 μm. **d** Inhibition of RhoGDIα phosphorylation by ZNF185 knockdown is not rescued by overexpression of N-terminal ZNF185. **e** N-terminal ZNF185 does not rescue stress fiber formation by ZNF185 knockdown. Immunofluorescence staining of actin (green) and VE-cadherin (magenta). Representative images are shown (*n* = 3). Scale bars, 25 μm.

resulting in an N-terminal actin-targeting domain and a C-terminal functional domain, which included a LIM domain and a PKA-interacting domain (Fig. 8b). The N-terminal ZNF185 lacking PKA-interacting domain did not rescue stress fiber formation by ZNF185 knockdown (Fig. 8e). The C-terminal ZNF185 changes intracellular localization and results in dispersion throughout the cytosol (Fig. 8c)[18], which suggests that C-terminal ZNF185 cleaved by thrombin is dysfunctional. Irreversible

dysfunction of ZNF185 may be a major cause of stress fiber formation by thrombin.

In conclusion, we demonstrated that ZNF185 is necessary for the maintenance of cytoskeletal actin structures in endothelial cells. ZNF185 interacts with actin, plectin, and PKA. ZNF185 knockdown suppresses cAMP/PKA/RhoA signaling and causes RhoA activation, stress fiber formation, and forskolin-resistant hyperpermeability.

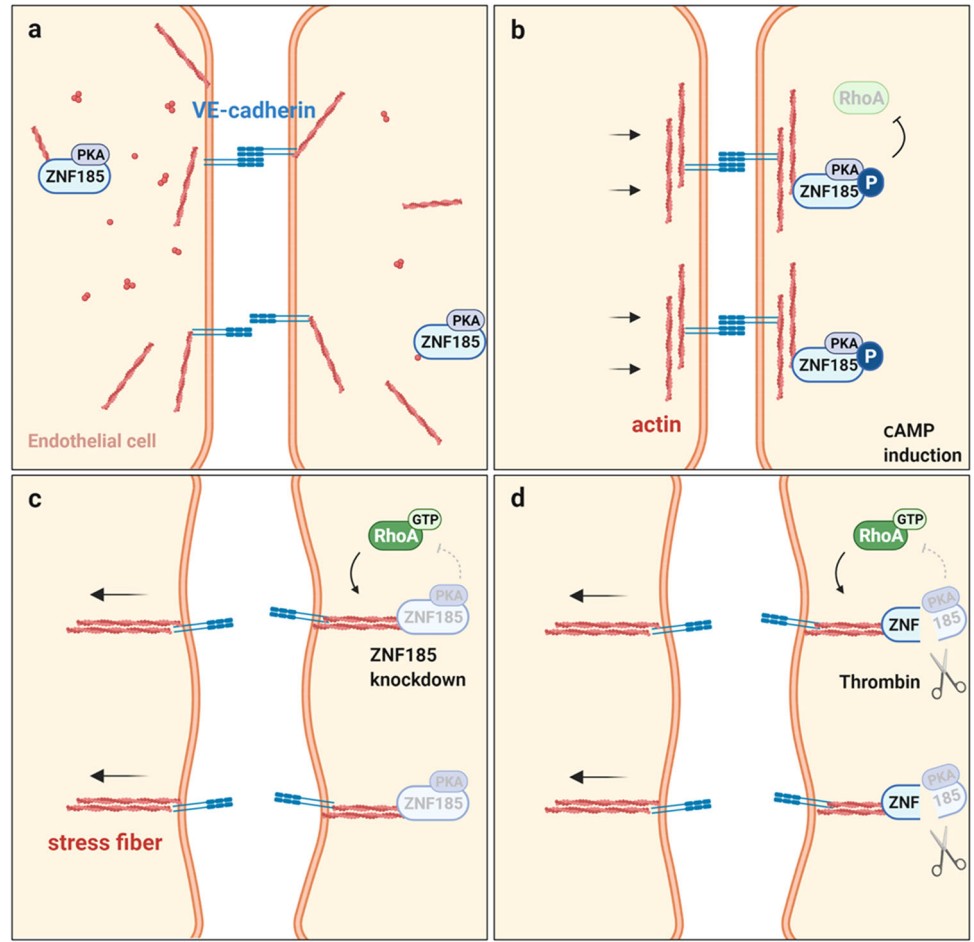

**Fig. 9 Schematic summary of the role of ZNF185 in endothelial barrier function. a** ZNF185 is localized to the perinuclear region under basal conditions in endothelial cells. N-terminal region of ZNF185 interacts with actin. **b** C-terminal region of ZNF185 interacts with PKA, which phosphorylates ZNF185. Both phospho-ZNF185 and F-actin accumulate at the membrane region in response to forskolin and stabilize cortical actin structures. **c** ZNF185 knockdown activates RhoA and promotes actin stress fiber formation, causing endothelial hyperpermeability. ZNF185 is essential for cAMP/PKA-induced RhoA inhibition. **d** The thrombin protease divides ZNF185 into an N-terminal actin-targeting domain and a C-terminal PKA-interacting domain, which may cause RhoA activation and stress fiber formation. Schematics were created with BioRender.com.

## Methods

**Study approvals.** All animal studies were performed in accordance with the guidelines for animal research of Tokyo Medical and Dental University. The study protocol was approved by the Animal Care and Use Committee of Tokyo Medical and Dental University (approval number: A2021-120C).

**Animals.** C57BL/6J mice were purchased from CLEA JAPAN (Tokyo, Japan). Male mice were used for experiments. *Zfp185* knockout mice were generated by Cyagen Biosciences, Inc (Suzhou, China) using the CRISPER/Cas9 genome-editing system. Six guide RNAs (gRNA1: 5′-GGT TTC CTT TCA GGT ATC AGG GG-3′, gRNA2: 5′-CAA GGC GAT GCT ACC CCC TAT GG-3′, gRNA3: 5′-TAA CGA TAC TGA AGT CGA TGT GG-3′, gRNA4: 5′-GCC TGG CCC CTA GTC AAC AAC GG-3′, gRNA5: 5′-TCT AAC GGG AAT GAA TAG GAT GG-3′, gRNA6: 5′-TAT TTC GCT CAA GTC AAG TTG GG-3′) were designed to target the region from exon 2 to exon 17 in mouse *Zfp185* gene. The mice carrying a deletion of 24794 bp in *Zfp185* gene were identified. This mutant strain was backcrossed to C57BL/6J for 10 generations to remove potential off-target variants. Genotype analysis was performed by Sanger sequencing using the following primers: *Zfp185*-Forward 1, 5′-CCT TCC CCA GAC TCT TTC CAT TAG-3′, *Zfp185*-Reverse 1, 5′-AAG ATT GTA GCT TCC ATG AGG CA-3′, *Zfp185*-Reverse 2, 5′-CAC CTT TCC CAA TAC TTC CTA A-3′. All mice were on the C57BL/6J background and were maintained under specific-pathogen-free conditions at 25 °C with a cycle of 12 h light/dark. Fresh water and rodent diet were available at all time.

**Cell culture and reagents.** HUVECs were obtained from PromoCell and cultured in 6 cm dishes using an endothelial growth medium (#C-22011; PromoCell GmbH). HEK 293T cells were cultured in 6 cm dishes in DMEM (Dulbecco's modified Eagle's medium) supplemented with 10% fetal bovine serum. Plasmid DNA was transfected into HEK293T cells using Lipofectamine 2000 reagent

(#10696153; Invitrogen Corporation, Carlsbad, CA). Forskolin (#F6886; Sigma–Aldrich Corporation, St. Louis, MO, USA), PKI 14-22 amide, myristoylated (#2546; Tocris Bioscience), Y-27632 (#08945-84; Nacalai Tesque, Inc. Kyoto, Japan), and thrombin (Fuji Pharma Corporation) were added to the cells for 1 h.

**Western blot analysis.** HUVECs and HEK293T cells were washed twice with phosphate-buffered saline (PBS) and solubilized in lysis buffer[44]. After centrifugation at 15,000 × *g* for 10 min at 4 °C, the protein concentration was measured by the Bradford protein assay (Expedeon Inc., San Diego, CA, USA). The supernatants were denatured in sodium dodecyl sulfate (SDS) sample buffer (Cosmo Bio, Co., Ltd., Tokyo, Japan) for 20 min at 60 °C. Whole homogenates of mouse ears were centrifuged for 10 min at 15,000 × *g*. Supernatants were denatured in SDS sample buffer as described above and were used to evaluate the expression of Zfp185. Equal amounts of protein were separated by SDS-PAGE, transferred to nitrocellulose membranes (GE Healthcare), and subjected to immunoblotting. Western blot analysis was performed with the following primary antibodies: mouse anti-β-actin (dilution, 1:1000; #A2228; Sigma–Aldrich Corporation), rabbit anti-β-actin (13E5) (dilution, 1:1000; #4970; Cell Signaling Technology, Inc., Beverly, MA, USA), mouse anti-GAPDH (dilution, 1:1000; #sc-32233; Santa Cruz), rabbit anti-phospho-PKA substrate (dilution, 1:1000; #9624; Cell Signaling Technology), mouse anti-Flag (M2) (dilution, 1:1000; #F3165, Sigma-Aldrich Corporation), mouse anti-Myc-Tag (9B11; Cell Signaling Technology), rabbit anti-Myc-tag (71D10) (dilution, 1:1000; #2278; Cell Signaling Technology), rabbit anti-HA-Tag (dilution, 1:1000; #C29F4; Cell Signaling Technology), rabbit anti-ZNF185 (dilution, 1:1000; #HPA000400; Sigma-Aldrich Corporation), rabbit anti-RhoA (dilution, 1:1000; #2117; Cell Signaling Technology), rabbit anti-RhoGDI (dilution, 1:1000; #ab133248; Abcam), rabbit anti-ARHGDIA (Phospho-Ser174) (dilution, 1:1000; #A1189; Assay Biotechnology Company), and rabbit anti-Plectin (dilution, 1:1000; #VPA00847; Bio-Rad Laboratories). Alkaline

phosphatase-conjugated anti-IgG (Promega Corporation) and goat anti-mouse light-chain antibody (dilution, 1:1000; #AP200A; Sigma-Aldrich) were used as the secondary antibody, and Western Blue (Promega Corporation) was used for signal detection. The band intensities of the Western blots were quantified using ImageJ software (https://imagej.nih.gov/ij/). Uncropped western blots are shown in Supplementary Figs. 13–21. All transferred membranes come from different gels, and each membrane is probed with only a single primary antibody. None of them are stripped and re-probed.

**Immunoprecipitation**. HUVEC and HEK293T cell lysates were incubated with anti-c-Myc Magnetic Beads (#88842, Pierce Biotechnology), anti-HA Magnetic Beads (#88837, Pierce Biotechnology), anti-FLAG M2 Affinity Gel (#A2220, Sigma-Aldrich Corporation), or rabbit anti-phospho-PKA substrate (Cell Signaling Technology) captured on protein G magnetic beads (Dynabeads® ProteinG, Invitrogen Corporation) for 1 h at 4 °C.

**Protein purification and mass spectrometry**. Forskolin or vehicle was applied to the HUVECs for 1 h. After treatment, the cell lysates were collected as described above. Anti-pPKA substrate antibody (Cell Signaling Technology) was bound to protein G magnetic beads (Invitrogen Corporation) and the cell lysates were purified according to the manufacturer's instructions. The proteins bound to the antibody were denatured in an SDS sample buffer (Cosmo Bio) for 20 min at 60 °C. Equal amounts of protein were separated by SDS-polyacrylamide gel electrophoresis and visualized by silver staining (Sil-Best Destain Kit, Nacalai Tesque). Three stained bands and the control bands shown in Supplementary Fig. 1a were excised from the gels. After the Myc-ZNF185 fragments shown in Fig. 2c were transfected into HEK293T cells, the lysates were collected, purified, and denatured as described above. Equal amounts of protein were separated by SDS-PAGE and visualized by silver staining (Nacalai Tesque). One stained band along with a control band (Supplementary Fig. 3a) was excised from the gels. LC-MS/MS analysis of the gel bands was conducted by MS BioWorks (Ann Arbor, MI, USA) using the following protocol. Each gel was washed with 25 mM ammonium bicarbonate followed by acetonitrile, reduced with 10 mM dithiothreitol at 60 °C, followed by alkylation with 50 mM iodoacetamide at room temperature. The proteins in the gels were digested with trypsin (Promega) at 37 °C for 4 h and quenched with formic acid. Half of each digested sample was analyzed by nano LC-MS/MS with a Waters NanoAcquity HPLC system interfaced to a ThermoFisher Fusion Lumos mass spectrometer. Peptides were loaded onto a trapping column and eluted over a 75 µm analytical column at 350 nL/min. Both columns were packed with Luna C18 resin (Phenomenex) and a 0.5 h gradient was applied. The mass spectrometer was operated in data-dependent mode with the Orbitrap operating at 60,000 FWHM and 15,000 FWHM for MS and MS/MS respectively. Advanced Peak Determination was enabled, and the instrument was run with a 3 s cycle time for MS and MS/MS. The data were analyzed using a local copy of Mascot (Matrix Science) with the following parameters: Enzyme: Trypsin/P; Database: SwissProt Mouse (concatenated forward and reverse plus common contaminants); Fixed modification: Carbamidomethyl (C); Variable modifications: Oxidation (M), Acetyl (N-term), Pyro-Glu (N-term Q), Deamidation (N/Q); Mass values: Monoisotopic; Peptide Mass Tolerance: 10 ppm; Fragment Mass Tolerance: 0.02 Da; and Max Missed Cleavages: 2. The Mascot DAT files were parsed into Scaffold (Proteome Software) for validation and filtering to create a nonredundant list per sample. Data were filtered using 1% protein and peptide FDR and requiring at least two unique peptides per protein.

**Immunofluorescent studies**. HUVECs were fixed with 4% paraformaldehyde and permeabilized with 0.1% Triton-X/PBS. Rabbit anti-phospho-PKA substrate (dilution, 1:200; #9624; Cell Signaling Technology), mouse anti-Myc-Tag (dilution, 1:200; #9B11; Cell Signaling Technology), rabbit anti-HA-tag (dilution, 1:200; #C29F4; Cell Signaling Technology), mouse anti-VE-cadherin (dilution, 1:200; #sc-9989; Santa Cruz), and rabbit anti-Halo-tag (dilution, 1:200; #G9281; Promega Corporation) were used as primary antibodies. Alexa 488 and 546 dye-labeled antibodies (dilution, 1:200; Molecular Probes) were used as the secondary antibodies. The actin cytoskeleton was detected using Acti-stain™ 555 phalloidin (dilution, 1:1000; #PHDH1-A; Cytoskeleton Inc.). The samples were mounted with ProLong™ Glass Antifade Mountant with NucBlue™ (P36981; Invitrogen Corporation). Immunofluorescence images were obtained using a Leica SP-8 confocal microscope. The intensity of stress fibers was quantified using ImageJ software (https://imagej.nih.gov/ij/).

**Lentiviral vector production and transduction**. N-terminal Myc-tagged and C-terminal HA-tagged ZNF185 were amplified from a ZNF185 (NM_007150) Human Tagged ORF Clone (#RC213230; ORIGINE) by PCR and cloned into the pENTR3C Dual Selection Vector (A10464; Invitrogen Corporation) using the Gibson assembly technique (New England Biolabs, Ipswich, MA, USA)[45]. N-terminal Myc-tagged plus C-terminal HA-tagged ZNF185 and pENTRegfp2 (Addgene, Cambridge, MA, USA #22450) were subsequently transferred into pLenti CMV Puro DEST (#17452; Addgene) using the LR reaction. Primer sequences used for PCR were summarized in Supplementary Table 1. In addition, the targeting sequences or nontargeting scramble control were cloned into the

pLKO.1 puro lentiviral plasmid. The target sequences are as follows: 5′-AGC CAG TAT CTG CAC GCT ATA-3′ for ZNF185, 5′-GGG TGT GGA GTA CCG GAT AAA-3′ for RhoGDIα, 5′-AGC CGC CAC TAA GGA GCT AAT-3′ for plectin. The recombinant lentiviral vectors, packaging vector psPAX2 (#12260; Addgene), and enveloping vector pMD2.G (#12259; Addgene) were cotransfected into HEK 293T cells. The culture medium was harvested and centrifuged at 500 × g for 5 min at 4 °C to remove any cellular debris. The supernatant containing 4 µg/ml polybrene (Sigma–Aldrich Corporation) was added to the HUVECs. The successful transduction was confirmed by identifying GFP-positive cells by fluorescence microscopy. For molecular analysis, mixed populations of cells without antibiotic selection were used. All experiments were conducted before three passages after transduction. Lentiviral transduction into HUVECs was performed repeatedly to obtain the experimental data.

**Permeability assay**. The numbers of wild-type HUVECs or HUVECs transduced with scrambled or ZNF185 shRNA were counted and equal numbers of HUVECs ($2 \times 10^6$ cells/ml) were seeded onto semipermeable filters (Transwell, 0.4 µm pore size; Corning Costar). HUVECs form mature monolayers in ~3 days. The passage of FITC-dextran (molecular mass 70 kDa; 10 mg/ml; #FD70; TdB) from the top to the bottom chamber was measured using a FilterMax™ F5 Multi-Mode Microplate Reader. FITC-dextran permeability was calculated as the ratio of measured experimental values to the control value.

**RhoA GTPase pulldown assay**. Rho activity in HUVECs was evaluated as follows: HUVEC lysates were incubated for 1 h at 4 °C with a glutathione S-transferase fusion protein containing the Rho binding domain of Rhotekin, which was previously captured on glutathione Sepharose beads (#17-0756-01; Cytiva). Lysates were then washed with PBS. GTP-binding Rho was released from the beads by an SDS sample buffer and the amount of GTP-binding Rho was determined using Western blot analysis using an anti-RhoA antibody as described above.

**In vivo vascular permeability assay**. Vascular permeability assays were performed[46]. Evans blue dye (30 mg/kg in 100 µl PBS) was injected into the tail vein. After 1 min, 5% mustard oil (#57-06-7; Tokyo Chemical Industry Co., Ltd.) diluted in mineral oil was applied to the dorsal and ventral surfaces of the ear with a cotton swabs. The pictures shown in Fig. 6 were captured 30 min after Evans blue injection. The Evans blue dye was extracted from the ears with 1 ml of formamide overnight at 55 °C and measured spectrophotometrically at 620 nm (FilterMax™ F5 Multi-Mode Microplate Reader). Vascular permeability was calculated by extraction of Evans blue dye and results are expressed as µg per tissue weight.

**Statistics and reproducibility**. Statistical significance was evaluated using an unpaired two-tailed Student's $t$ test. Data are presented as the mean ± standard error. A probability value of <0.05 was considered statistically significant. At least three independent experiments were performed to ensure reproducibility. Sample sizes and the number of replicates are described in the respective figure legend.

**Reporting summary**. Further information on research design is available in the Nature Portfolio Reporting Summary linked to this article.

## Data availability
The data that support the findings of this study are available from the corresponding author upon reasonable request. The numerical source data for graphs and charts are available in Supplementary Data 1.

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

## Acknowledgements

This work was supported by Grants-in-Aid for Scientific Research (B) (21H02933 to F.A.) and Scientific Research (A) (19H01049 to S.U.) from the Japan Society for the Promotion of Science, a Health Labour Science Research Grant from the Ministry of Health Labour and Welfare, AMED under Grant Number JP19ek0109304 and JP19lm0203023 to S.U., TMDU priority research areas grant to F.A., TMDU Young Innovative Medical Scientist Unit to F.A., The Uehara Memorial Foundation to F.A., Takeda Science Foundation to F.A., Pharmacodynamics Research Foundation to F.A., Japan Intractable Diseases (Nanbyo) Research Foundation (2018A01) to F.A., MSD Life Science Foundation, Public Interest Incorporated Foundation to F.A.

## Author contributions

S.S. and F.A. performed, analyzed, and interpreted the experiments and wrote the paper. S.K., Y.H., T.F., S.M., K.S., T.M., E.S., T.R., and S.U. analyzed and interpreted the data. F.A. supervised the project and designed all the experiments.

## Competing interests

The authors declare no competing interests.
