## [Peer Review File · Communications Biology]

Reviewers' comments:

Reviewer #1 (Remarks to the Author):

The manuscript from Suzuki et al. shows a nice description of a new interaction between PKA and the actin-associated protein ZNF185 (Zhang, *Oncogene*, 2007). It also describes the effect of ZNF185 on basal and PKA-induced barrier function and F-actin remodeling, whereas it concludes with the finding that thrombin cleaves ZNF185.

The impeccable description of the interaction is followed by a superficial analysis of the effect of this protein in the regulation of stress fibers and RhoA activation. Why is ZNF185 activating RhoA?. Is it a ZNF185-mediated signaling process or ZNF185 plays a structural role in organizing actin filaments?. How is the signaling downstream PKA in ZNF18-depleted cells?. What is the effect of expressing the ZNF185 mutant lacking the PKA-phosphorylated residues on stress fibers and Rho activation?.

Thrombin, which is shown as an unrelated positive control in the permeability assays in Figure 4, surprisingly takes central stage in the last figure containing experimental data. Thrombin cleaves ZNF185. This does not seem very connected to the previous experiments, particularly when the manuscript abruptly finishes with this set of data and does not address whether ZNF185 cleavage mediates thrombin effect on barrier function or whether ZNF185 cleavage has anything to do with PKA-mediated signaling. Overall the manuscript elegantly describes a new interaction but it performs a superficial and disorganized functional characterization of the role of this protein which, in some cases, such as in Figure 5, is not connected to the initial characterization of ZNF185 regulation by PKA.

Minor point. Is it difficult to believe that all pPKA signal at cell-cell junctions "almost disappeared" upon shRNA-mediated knock down of ZNF185 in Figure 1f. The images do not show so, the experiment is not quantified and Figure 3d clearly shows that cell-cell junctions became disrupted by the absence of ZNF185, which obviously reduced the staining of any antibody on these cellular domains. The text equals pPKA antibody signal to pZNF185 levels, without enough evidence. This experiment should be eliminated or at least pPKA levels at junctions quantified and the text rewritten.

Reviewer #2 (Remarks to the Author):

In the present manuscript the authors identified ZNF185 as a novel PKA substrate and demonstrate that it is required for the organization of endothelial cell-cell junctions and the suppression of vascular hyperpermeability by regulating the cAMP/PKA/RhoA signaling pathway. Overall, it is an interesting story, with rationally designed and well-performed experiments. Below are some stipulations, by the order they were identified, to help the authors improve their manuscript:

- In Fig. 1 treatment with Forskolin for 1 h was identified via time course or references? Normally less time is required.
- In Fig. 1C under no transfection conditions, a ZNF185 band should be present in the WB (in lower molecular weight) upon development with the ZNF185 Ab. Did the authors detect it or was it too faint?
- Fig. 1F does not include plectin knockdown, as mentioned in lines 230-1.
- In Fig. 2A, where does the pZNF185 quantification correspond to? Also, in the same figure, it is good to mention the molecular weight of the bands in the HA development.
- In Fig. 2D, what does the strong band in the Flag development of the IP:Myc correspond to (since it is less than 50 kD)?
- In Fig. 2E, what about RIIb? Does it interact with Δ PKA?
- What is the impact of ZNF185- Δ PKA in stress fiber formation?
- There is no direct proof (i.e. staining for plectin) that plectin is included in the complex in Figs. 3d,e and Fig. 4a,b, as it is claimed in the discussion.
- In Fig. 4B what does Sc and Sh stand for? I guess Control and Sh treatment respectively, but it

is not mentioned in the text (results section) or the Figure legend.

- In Fig. 4E, what is the outcome of Fsk or Y-27632 treatments in the presence ZNF185? Inclusion of these groups would be helpful.
- In the figure legends the authors should be consistent by using either parentheses or not in all the letters. As an example, in Fig 5, a and b are in parentheses while the rest of the letters are not.
- The logical question that comes up would be the overexpression of the two “cleaved” fragments (Myc- and HA-tagged) as recombinant proteins, to identify the efficiency of each one to activate stress fibers and compare it with the ZNF knockdown.
- An in vivo permeability experiment, such as Miles assay, with mutant ZNF185 or the recombinantly expressed ZNF185 fragments upon thrombin cleavage would provide a more preclinical impact of the present findings.
- In the materials and methods, some antibodies are missing cat nos.

Reviewer #3 (Remarks to the Author):

The study by Suzuki et al focuses on cAMP/protein kinase A (PKA) signaling in promoting endothelial barrier function. Previous studies agree that cAMP/PKA are major effectors of vascular permeability and protect the endothelial barrier; however, the molecular mechanisms underlying local PKA networks in endothelial cells remain undefined.

Here, the authors have identified ZNF185 binds to PKA and is a PKA substrate. This interaction is shown to be essential in regulating cAMP/PKA signaling and maintain integrity of the endothelial barrier. Results demonstrate that ZNF185 co-localises with the membranous actin cytoskeleton upon cAMP activation. ZNF185 knockdown increases endothelial barrier permeability and actin stress fiber formation through constitutive RhoA activation, indicating ZNF185 to be necessary for physiological endothelial mechanics. Further work showed that ZNF185 is cleaved by thrombin, a known promoter of stress fiber formation, and it is suggested that ZNF185 cleavage might contribute to thrombin induced barrier breakdown. Together, this manuscript identifies ZNF185 as a novel PKA substrate and subcellular localisation-determining PKA-interacting protein, necessary for maintenance of cytoskeletal actin structures in endothelial cells.

The findings are very interesting and I suggest that the work is suited for publication in Communications Biology once the following comments have been addressed.

Major comments:

1. How were the fragments of Myc-tagged ZNF185 in Fig 2c generated? It would be helpful to have a description of ZNF185 protein domains and which of these are included in the distinct fragments. It is discussed in the introduction that ZNF185 has an N-terminal actin-binding domain and a C-terminal LIM protein domain to mediate protein-protein interactions. Where are these in fragments A-D?
2. In Fig 2d (right panel), there are bands showing up in control lanes of the IP: Myc gel that are not present in the INPUT. I.e. the Myc antibody has positive bands in non-Myc-ZNF185 stimulated cell lanes, and the Flag antibody similarly has positive bands in the non-PKA-Flag stimulated cell lanes. Can the authors clarify what are the exact bands the reader should focus on in the IP?
3. In Fig 3d and e the authors examine the effect of ZNF185 loss on stress fibers and VE-cadherin in an endothelial monolayer. Can the authors explain what the take-home message is from this same experiment on single cells (Fig 3b)? To me the monolayer seems more informative in terms of permeability so perhaps the single cell data can be moved to supplement?
4. Fig 3d and e. Fig 3d shows changes in VE-cadherin and F-actin expression in the monolayer. The resolution of this data needs to be improved. For example, the arrows indicate zig-zag junctions and the arrowheads gaps. This is difficult to see in these images and therefore I suggest adding high res, high-magnification data of these junctional phenotypes. Fig 3 e shows the distribution of F-actin along cell-cell junctions. Although this is visually a really nice way to show that junctions are either zig-zag (irregular) or straight, this data is not quantitative as suggested in the Figure legend. To show whether there are changes in stress fiber

distribution the authors really need to quantify F-actin expression (junctional vs non-junctional) across multiple junctions from multiple experiments. This will allow statistical testing of this phenotype between the different conditions.

I agree with the authors that there is a qualitative difference in stress fibers when comparing Sh-ZNF185 versus Sh-scramble. This is remarkably improved with forskolin treatment in the images, which does not seem as clearly reflected in panel e. Again, highlighting the need for more extensive quantitative analysis of this data.

A final suggestion related to this section of the story is for the authors to elaborate more on VE-cadherin junctional integrity in the main text, which focusses almost exclusively on the actin, yet effects on VE-cadherin are prominent (at least qualitatively).

5. In Fig 4d-e, what is the effect of RhoA inactivation (Y27632 treatment) on Sh-scramble control cells? This control should be included and quantified.

6. What is the recombinant ZNF185 referred to in Fig 5? It is unclear if this is referencing the C-terminal mutant version of ZNF185 shown in Supplement Fig 2 or a different recombinant version of the protein.

7. Finally, in Fig 5 and associated text, the authors mention that cleavage of ZNF185 by thrombin may impair ZNF185 function similar to that of ZNF185 knockdown, causing RhoA activation and stress fibers. The authors could substantiate this hypothesis nicely by investigating whether expression of a ZNF185 variant that lacks the thrombin cleavage site in ZNF185 knockdown cells can rescue the effects of thrombin of EC junctions.

Minor comments:

1. In Fig 1d, immunofluorescence showed localisation of Myc-tagged ZNF185 with pPKA substrates to the membrane region upon forskolin treatment. These images are convincing, but the observations would be strengthened by higher resolution images with a counterstain for VE-cadherin or a membrane marker.

2. In Fig 2a, it might be helpful to indicate which pPKA substrates band is ZNF185. From the IP:HA gel it is clear that this is the ~90kDa band, but the multiple bands in the INPUT gel can be distracting without an arrowhead or asterisk at the bands of interest?

3. In Fig 3a, the bands slightly larger than actin appear in the IP : HA gel and can be quite distracting from the fainter actin bands ~37kDa. As I understand it, the purpose of these results is to show that with ZNF185 overexpression and with forskolin treatment, actin is localised to ZNF185. So, what are these dark bands and could these be removed? It also would be helpful if the actin levels here were quantified similar to in Fig 1c or 2a to clearly demonstrate how this is changing with forskolin treatment.

4. Information on the sh-ZNF185 knockdown and associated scrambled sequences is missing from the methods section.

Point-by-point reply to reviewers.

We thank the reviewers for their thoughtful comments, which have helped to improve the manuscript. We feel that we could successfully address most of the previously raised concerns. Our replies to the reviewers' comments are below.

Reviewer #1 :

The manuscript from Suzuki et al. shows a nice description of a new interaction between PKA and the actin-associated protein ZNF185 (Zhang, *Oncogene*, 2007). It also describes the effect of ZNF185 on basal and PKA-induced barrier function and F-actin remodeling, whereas it concludes with the finding that thrombin cleaves ZNF185.

1. The impeccable description of the interaction is followed by a superficial analysis of the effect of this protein in the regulation of stress fibers and RhoA activation. Why is ZNF185 activating RhoA?. Is it a ZNF185-mediated signaling process or ZNF185 plays an structural role in organizing actin filaments?. How is the signaling downstream PKA in ZNF18-depleted cells?. What is the effect of expressing the ZNF185 mutant lacking the PKA-phosphorylated residues on stress fibers and Rho activation?.

Response:

We wish to express our strong appreciation to the reviewer for insightful comments on our paper. It has been reported that RhoGDI α mediates PKA/RhoA/actin stress fibers signaling pathway. DerMardirossian *et al.* and Qiao *et al.* demonstrated that PKA phosphorylates RhoGDI α at S174 (PMID: 15225553, 18768928). Phosphorylation of RhoGDI α -S174 increases the binding affinity between RhoGDI α and RhoA and then sequesters RhoA at the plasma membrane to suppress RhoA activity (PMID: 23012358). RhoA is conversely activated by *RhoGDI α* knockout, leading to elevation of basal endothelial permeability *in vivo* (PMID: 17525371). Additionally, knockdown of RhoGDI α in endothelial cell lines promotes stress fiber formation and disrupts adherens junctions (PMID: 17525371).

In line with previous reports, we confirmed that knockdown of RhoGDI α by sh-RhoGDI α induced RhoA activation and stress fiber formation in HUVECs, as shown in figure below. PKA activation by forskolin did not improve RhoGDI α knockdown-induced stress fiber formation, indicating that RhoGDI α was involved in PKA/RhoA/actin stress fibers signaling pathway.

We next examined the effects of ZNF185 knockdown on phosphorylation of RhoGDI α -S174 because ZNF185 interacted with PKA (Fig. 2b-e), and RhoGDI α -S174 is phosphorylated by PKA (PMID: 15225553, 18768928). Although forskolin phosphorylated pRhoGDI α -S174 in sh-scrambled HUVECs, its phosphorylation levels were consistently suppressed by ZNF185 knockdown regardless of the administration of forskolin. These results indicate that ZNF185 mediated PKA-induced phosphorylation of RhoGDI α -S174, leading to inhibition of RhoA and stress fiber formation.

We made a schematic signaling pathway of cAMP/PKA/RhoA in Supplementary Figure 10.

We added “RhoGDI α is a well-known mediator of PKA/RhoA signaling pathway. DerMardirossian *et al.* and Qiao *et al.* demonstrated that PKA phosphorylates RhoGDI α at S174^{29,30}, which increases the binding affinity between RhoGDI α and RhoA and then sequesters RhoA to suppress RhoA activity³¹. RhoA is conversely activated by *RhoGDI α* knockout, leading to elevation of basal endothelial permeability *in vivo*³². Additionally, knockdown of RhoGDI α in endothelial cell lines disrupts adherens junctions and promotes stress fiber formation³². In line with previous reports, we confirmed that knockdown of RhoGDI α by sh-RhoGDI α activated RhoA (Fig. 5a) and induced stress fiber formation in HUVECs (Fig. 5b-d). PKA activation by forskolin did not improve stress fiber formation, indicating that RhoGDI α was an indispensable mediator of PKA/RhoA/actin stress fibers signaling pathway. We next examined the effects of ZNF185 knockdown on phosphorylation of RhoGDI α -S174 because ZNF185

interacted with PKA (Fig. 2b-e), and RhoGDI α -S174 is phosphorylated by PKA^{29,30}. Although forskolin phosphorylated RhoGDI α -S174 in sh-scrambled HUVECs, its phosphorylation levels were consistently suppressed by ZNF185 knockdown regardless of the administration of forskolin (Fig. 5e). These results indicate that ZNF185 was essential for PKA-induced phosphorylation of RhoGDI α -S174, leading to inhibition of RhoA activity and stress fiber formation.” in p.11.

2. Thrombin, which is shown as an unrelated positive control in the permeability assays in Figure 4, surprisingly takes central stage in the last figure containing experimental data. Thrombin cleaves ZNF185. This does not seem very connected to the previous experiments, particularly when the manuscript abruptly finishes with this set of data and does not address whether ZNF185 cleavage mediates thrombin effect on barrier function or whether ZNF185 cleavage has anything to do with PKA-mediated signaling. Overall the manuscript elegantly describes a new interaction but it performs a superficial and disorganized functional characterization of the role of this protein which, in some cases, such as in Figure 5, is not connected to the initial characterization of ZNF185 regulation by PKA.

Response:

We strongly appreciate the reviewer's comment on this point. Fragments of ZNF185 cleaved by thrombin were helpful in understanding physiological roles of ZNF185. We first examined whether the cleavage of ZNF185 caused thrombin-induced endothelial barrier dysfunction. We generated stable cell lines of HUVECs overexpressing full-length ZNF185 or ZNF185- Δ exon 11,12 (without cleavage site). As shown in figure below, although thrombin induced stress fiber formation and disruption of adherens junctions (indicated by white arrowheads) in HUVECs overexpressing full-length ZNF185, ZNF185- Δ exon 11,12 rescued these disruptive effects of thrombin,.

Thrombin divided ZNF185 into an N-terminal actin-targeting domain and a C-terminal PKA-interacting domain. Importantly, both fragments lost their physiological function. We next overexpressed N-terminal ZNF185 (aa 1-245) and C-terminal ZNF185 (aa 305-689) in HUVECs, as shown in figure below. Consistent with the results of full-length ZNF185 (Fig. 1e), N-terminal ZNF185 (green) was translocated to the membrane region in response to forskolin. By contrast, C-terminal ZNF185 (magenta) was dispersed throughout the cytosol and did not respond to forskolin. These results indicate that cleaved C-terminal fragment of ZNF185 was mislocalized by thrombin.

Intracellular localization of N-terminal ZNF185 was not impaired. We thus examined whether N-terminal ZNF185 lacking PKA-interacting domain was sufficient to inhibit stress fiber formation. Overexpression of N-terminal ZNF185 lacking PKA-interacting domain was sufficient to inhibit stress fiber formation. Overexpression of N-terminal ZNF185 with ZNF185 knockdown failed to phosphorylate RhoGDI α -S174, as shown in figure below (Left). As a result, sh-ZNF185-induced stress fiber formation was not rescued by overexpression of N-terminal ZNF185 (Right). These results indicate that both N- and C-terminal ZNF185 were indispensable for maintaining physiological function of ZNF185.

We added “To investigate the role of ZNF185 cleavage in HUVECs, we generated stable cell lines overexpressing full-length ZNF185 or ZNF185- Δ exon 11,12. Although thrombin induced stress fiber formation and disruption of endothelial adherens junctions in HUVECs overexpressing full-length ZNF185, ZNF185- Δ exon 11,12 rescued these disruptive effects of thrombin (Fig. 8a, Supplementary Fig. 9a, b).

ZNF185 was irreversibly divided into an N-terminal actin-targeting domain and a C-terminal PKA-interacting domain by thrombin. Importantly, both fragments lost their physiological function. We overexpressed N-terminal ZNF185 (aa 1-245) and C-terminal ZNF185 (aa 305-689) in HUVECs (Fig. 8b). Consistent with the results of full-length ZNF185 (Fig. 1e), N-terminal ZNF185 (green) was translocated to the membrane region in response to forskolin (Fig. 8c). By contrast, C-terminal ZNF185 (magenta) was dispersed throughout the cytosol and did not respond to forskolin. These results indicate that cleaved C-terminal fragment of ZNF185 was mislocalized by thrombin.

Intracellular localization of N-terminal ZNF185 was not impaired. We next examined whether N-terminal

ZNF185 lacking PKA-interacting domain was sufficient to inhibit stress fiber formation. Overexpression of N-terminal ZNF185 with ZNF185 knockdown failed to phosphorylate RhoGDI α -S174 (Fig. 8d). As a result, ZNF185 knockdown-induced stress fiber formation was not rescued by N-terminal ZNF185 (Fig. 8e, Supplementary Fig. 9c, d). These results indicate that both N- and C-terminal ZNF185 were indispensable for maintaining physiological function of ZNF185. Thrombin-induced irreversible cleavage of ZNF185 was one of the causes of stress fiber formation in HUVECs.” in p.13.

3. Minor point. Is it difficult to believe that all pPKA signal at cell-cell junctions “almost disappeared” upon shRNA-mediated knock down of ZNF185 in Figure 1f. The images do not show so, the experiment is not quantified and Figure 3d clearly shows that cell-cell junctions became disrupted by the absence of ZNF185, which obviously reduced the staining of any antibody un these cellular domains. The text equals pPKA antibody signal to pZNF185 levels, without enough evidence. This experiment should be eliminated or at least pPKA levels at junctions quantified and the text rewritten.

Response:

We wish to thank the reviewer for this comment. In accordance with the reviewer's comment, we eliminated Figure 1f.

Reviewer #2 :

In the present manuscript the authors identified ZNF185 as a novel PKA substrate and demonstrate that it is required for the organization of endothelial cell–cell junctions and the suppression of vascular hyperpermeability by regulating the cAMP/PKA/RhoA signaling pathway. Overall, it is an interesting story, with rationally designed and well-performed experiments. Below are some stipulations, by the order they were identified, to help the authors improve their manuscript:

1. In Fig. 1 treatment with Forskolin for 1 h was identified via time course or references? Normally less time is required.

Response:

We thank the reviewer's important comments. ZNF185 was rapidly phosphorylated by forskolin. The maximal effect was achieved within 5 minutes.

We added "Forskolin rapidly phosphorylated ZNF185 within 5 minutes (Fig. 1d)." in p.7.

2. In Fig. 1C under no transfection conditions, a ZNF185 band should be present in the WB (in lower molecular weight) upon development with the ZNF185 Ab. Did the authors detect it or was it too faint?

Response:

We would like to thank the reviewer for valuable comments. We confirmed that ZNF185 Ab detected endogenous levels of ZNF185 in HUVEC cell lysate using knockdown experiments (Supplementary Fig.3). As shown in figure below, we added a black arrow which indicated endogenous ZNF185 bands and added arrows which indicated overexpressed ZNF185 bands.

We added "An anti-ZNF185 antibody detected endogenous ZNF185 indicated by a black arrowhead as well as overexpressed ZNF185 indicated by arrows (Fig. 1c). The band intensity of endogenous ZNF185 was

reduced by ZNF185 knockdown (Supplementary Fig.3)." in p.7.

3. Fig. 1F does not include plectin knockdown, as mentioned in lines 230-1.

Response:

We thank you for your thoughtful comments. In accordance with reviewer 1's comment, we deleted Fig. 1f. Instead, we added plectin knockdown in supplementary Fig. 12. Plectin knockdown induced stress fiber formation, as shown in figure below. Furthermore, phosphorylation of PKA substrates at the membrane region indicated by white arrows were markedly suppressed by knockdown of plectin or ZNF185 (indicated by white arrowheads).

We added "The functional role of plectin was quite similar to that of ZNF185. Knockdown of plectin also induced stress fiber formation in HUVECs (Supplementary Fig. 12a, b). Furthermore, phosphorylation of PKA substrates at the membrane region indicated by white arrows were markedly suppressed by knockdown of plectin or ZNF185 (indicated by white arrowheads) (Supplementary Fig. 12c)." in p.16.

4. In Fig.2A, where does the pZNF185 quantification correspond to? Also, in the same figure, it is good to mention the molecular weight of the bands in the HA development.

Response:

We greatly appreciate the reviewer's comment. We added black and red arrowheads which indicate bands of ZNF185. Band intensities of pZNF185 indicated by a red arrowhead were quantified. We also added the molecular weight of the bands in the HA development.

We added “As shown in Fig. 2a, forskolin-induced phosphorylation of ZNF185 indicated by arrowheads was attenuated by the PKA inhibitor, Myr-PKI, confirming that ZNF185 was phosphorylated by PKA.” In p.7.

5. In Fig. 2D, what does the strong band in the Flag development of the IP:Myc correspond to (since it is less than 50 kD)?

Response:

We thank the reviewer’s important comments. As the reviewer pointed out, the strong bands in the Flag development of the IP:Myc were quite confusing for readers. To delete IgG heavy-chain bands at 50 kDa, we used goat anti-mouse light-chain antibody (AP200A, Sigma-Aldrich) as the secondary antibody. Light-chain bands in the IP:Myc development are indicated by a black arrow. Band intensities of Flag (PKA RII α -ZNF185 interaction) indicated by a red arrowhead were quantified.

6. In Fig. 2E, what about RIIb? Does it interact with Δ PKA?

Response:

We wish to thank the reviewer for this comment. Similar to PKA RII α , PKA RII β did not interact with ZNF185- Δ PKA, as shown in figure below.

We added ” Unlike the full-length ZNF185, neither PKA RII α nor PKA RII β interacted with the ZNF185 deletion mutant (ZNF185- Δ PKA) (Fig. 2e, Supplementary Fig. 4).” In p.8.

7. What is the impact of ZNF185- Δ PKA in stress fiber formation?

Response:

We thank the reviewer for this comment. To examine the impact of ZNF185- Δ PKA in stress fiber formation, knockdown or knockout of endogenous *ZNF185* is required; however, ZNF185 knockdown causes severe stress fiber formation in HUVECs and impairs normal cell growth and proliferation, as shown in figure below. Therefore, it is difficult to precisely evaluate the role of ZNF185- Δ PKA in stress fiber formation.

Calcein AM Cell Viability Assay

We generated HUVECs overexpressing Myc-ZNF185-HA or Myc-ZNF185- Δ PKA-HA. However, Myc-ZNF185- Δ PKA-HA did not induce stress fiber formation, as shown in figure below. The endogenous ZNF185 may contribute to prevention of stress fiber formation.

8. There is no direct proof (i.e. staining for plectin) that plectin is included in the complex in Figs. 3d,e and Fig. 4a,b, as it is claimed in the discussion.

Response:

The reviewer's comment is correct. We tried to stain plectin in HUVECs; however, commercially available antibodies we used (rabbit anti-Plectin (#VPA00847; Bio-Rad Laboratories), rabbit anti-Plectin-1 (#2863; Cell Signaling Technology), and mouse anti-Plectin (#AP200A; Santa Cruz) could not detect endogenous plectin in HUVECs. In addition, plectin is a high molecular weight protein (400-500 kDa), and it is quite difficult to overexpress plectin in HUVECs.

We thus double-stained plectin and ZNF185 in HEK293T cells overexpressing Halo-plectin and Myc-ZNF185. Halo (plectin) and Myc (ZNF185) were expectedly colocalized at the membrane region, as shown in figure below.

We added "Overexpressed plectin and ZNF185 were well colocalized at the membrane region (Supplementary Fig.11e)" in p.16.

9. In Fig. 4B what does Sc and Sh stand for? I guess Control and Sh treatment respectively, but it is not mentioned in the text (results section) or the Figure legend.

Response:

We thank the reviewer's important comments. We replaced Sc with Sh-Scramble and replaced Sh with Sh-ZNF185. In addition, we added "After knockdown of ZNF185 by Sh-ZNF185, 10 μM forskolin or 10 U/ml thrombin was administered to HUVECs for 1 h." in figure legends"

10. In Fig. 4E, what is the outcome of Fsk or Y-27632 treatments in the presence ZNF185? Inclusion of these groups would be helpful.

Response:

We thank you for your thoughtful comments. The effects of Fsk on endothelial permeability in WT-HUVECs (Fig.4a) and Sh-scrambled HUVECs were the same, as shown in figure below. Y-27632 has been reported to decrease endothelial permeability (PMID: 18097057, PMID: 28316141). In line with these reports, Y27632 reduced endothelial permeability in Sh-scrambled control cells.

We added "Inhibition of RhoA/Rho-associated protein kinase signaling by Y-27632 inhibited baseline stress fiber assembly (Fig. 4d, Supplementary Fig. 7a, b) and endothelial permeability (Fig. 4e) in sh-scrambled HUVECs, as previously reported^{27,28}. Y-27632 successfully counteracted stress fiber formation via ZNF185 knockdown (Fig. 4d) and potently ameliorated forskolin-resistant hyperpermeability caused by ZNF185 knockdown (Fig. 4e)." in p.11.

11. In the figure legends the authors should be consistent by using either parentheses or not in all the letters. As an example, in Fig 5, a and b are in parentheses while the rest of the letters are not.

Response:

As per the reviewer's comment, we replaced (a) with a. and replaced (b) with b. in the figure legends.

12. The logical question that comes up would be the overexpression of the two "cleaved" fragments (Myc- and HA-tagged) as recombinant proteins, to identify the efficiency of each one to activate stress fibers and compare it with the ZNF knockdown.

Response:

We greatly appreciate the reviewer's comment. Thrombin divided ZNF185 into an N-terminal actin-targeting domain and a C-terminal PKA-interacting domain. Both fragments expectedly lost their physiological function. We overexpressed N-terminal ZNF185 (aa 1-245) and C-terminal ZNF185 (aa 305-689) in HUVECs, as shown in figure below. Consistent with the results of full-length ZNF185 (Fig. 1e), N-terminal ZNF185 (green) was translocated to the membrane region in response to forskolin. By contrast, C-terminal ZNF185 (magenta) was dispersed throughout the cytosol and did not respond to forskolin. These results indicate that cleaved C-terminal fragment of ZNF185 was mislocalized by thrombin.

Intracellular localization of N-terminal ZNF185 was not impaired. We next examined whether N-terminal ZNF185 lacking PKA-interacting domain was sufficient to inhibit stress fiber formation. The target sequence of sh-ZNF185 plasmid is in the C-terminus of ZNF185. Thus, we simultaneously transfected sh-ZNF185 and Myc-N-terminal ZNF185. Overexpression of N-terminal ZNF185 failed to rescue stress fiber formation by ZNF185 knockdown, as shown in figure below. These results indicate that both N- and C-terminal ZNF185 were indispensable for maintaining physiological function of ZNF185.

We added “ZNF185 was irreversibly divided into an N-terminal actin-targeting domain and a C-terminal PKA-interacting domain by thrombin. Importantly, both fragments lost their physiological function. We overexpressed N-terminal ZNF185 (aa 1-245) and C-terminal ZNF185 (aa 305-689) in HUVECs (Fig. 8b). Consistent with the results of full-length ZNF185 (Fig. 1e), N-terminal ZNF185 (green) was translocated to the membrane region in response to forskolin (Fig. 8c). By contrast, C-terminal ZNF185 (magenta) was dispersed throughout the cytosol and did not respond to forskolin. These results indicate that cleaved C-terminal fragment of ZNF185 was mislocalized by thrombin.

Intracellular localization of N-terminal ZNF185 was not impaired. We next examined whether N-terminal ZNF185 lacking PKA-interacting domain was sufficient to inhibit stress fiber formation. Overexpression of N-terminal ZNF185 with ZNF185 knockdown failed to phosphorylate RhoGDI α -S174 (Fig. 8d). As a result, ZNF185 knockdown-induced stress fiber formation was not rescued by N-terminal ZNF185 (Fig. 8e, Supplementary Fig. 9c, d). These results indicate that both N- and C-terminal ZNF185 were indispensable for maintaining physiological function of ZNF185. Thrombin-induced irreversible cleavage of ZNF185 was one of the causes of stress fiber formation in HUVECs.” in p.14.

13. An in vivo permeability experiment, such as Miles assay, with mutant ZNF185 or the recombinantly expressed ZNF185 fragments upon thrombin cleavage would provide a more preclinical impact of the present findings.

Response:

We wish to express our strong appreciation to the reviewer for insightful comments. Mouse *Zfp185* is an orthologous gene of human *ZNF185*. Similar to ZNF185, *Zfp185* interacted with PKA and actin, as shown in figure below. We then generated *Zfp185* knockout mice using CRISPR/Cas9 genome editing system. Evans blue dye was used to measure vascular permeability. We measured the vascular permeability response to application of mustard oil to the right ears of mice. *Zfp185* knockout significantly increased vascular permeability.

In contrast to ZNF185, Zfp185 does not contain the thrombin-susceptible sequence, LTPRAGLR. In our future studies, we would like to examine the effects of thrombin on ZNF185 using human tissues.

We added “Mouse *Zfp185* is an orthologous gene of human *ZNF185*³³. Similar to ZNF185, Zfp185 interacted with PKA subunits (RIα, RIβ, RIIα, and RIIβ) as well as actin (Fig. 6a, b). We then generated *Zfp185* knockout mice using CRISPR/Cas9 genome editing system (Fig. 6c-e, Supplementary Fig. 8). Evans blue dye was used to measure vascular permeability. The topical application of mustard oil to the right ear, which induces cutaneous inflammation, enhanced Evans blue dye leakage in *Zfp185* knockout mice (Fig. 6f, g). These results indicate that Zfp185 is involved in vascular permeability under inflammatory conditions.” in p.12.

14. In the materials and methods, some antibodies are missing cat nos.

Response:

We appreciate the reviewer’s comment. We added the catalog numbers in the materials and methods.

Reviewer #3 :

The study by Suzuki et al focuses on cAMP/protein kinase A (PKA) signaling in promoting endothelial barrier function. Previous studies agree that cAMP/PKA are major effectors of vascular permeability and protect the endothelial barrier; however, the molecular mechanisms underlying local PKA networks in endothelial cells remain undefined.

Here, the authors have identified ZNF185 binds to PKA and is a PKA substrate. This interaction is shown to be essential in regulating cAMP/PKA signaling and maintain integrity of the endothelial barrier. Results demonstrate that ZNF185 co-localises with the membranous actin cytoskeleton upon cAMP activation. ZNF185 knockdown increases endothelial barrier permeability and actin stress fiber formation through constitutive RhoA activation, indicating ZNF185 to be necessary for physiological endothelial mechanics. Further work showed that ZNF185 is cleaved by thrombin, a known promoter of stress fiber formation, and it is suggested that ZNF185 cleavage might contribute to thrombin induced barrier breakdown. Together, this manuscript identifies ZNF185 as a novel PKA substrate and subcellular localisation-determining PKA-interacting protein, necessary for maintenance of cytoskeletal actin structures in endothelial cells.

The findings are very interesting and I suggest that the work is suited for publication in Communications Biology once the following comments have been addressed.

Major comments:

1. How were the fragments of Myc-tagged ZNF185 in Fig 2c generated? It would be helpful to have a description of ZNF185 protein domains and which of these are included in the distinct fragments. It is discussed in the introduction that ZNF185 has an N-terminal actin-binding domain and a C-terminal LIM protein domain to mediate protein-protein interactions. Where are these in fragments A-D?

Response:

We greatly appreciate the reviewer's comment. Actin binding domain lies within the NH₂-terminal 135 a.a. residues (ref.18). ZNF185 contains a single LIM domain in COOH-terminus (ref.18). We added a description of ZNF185 protein domains in Fig. 2c. In addition, we confirmed that Myc-tagged ZNF185 fragment A bound to actin (indicated by red arrow in Fig. 2d).

We added "In line with previous reports¹⁸, actin interacted with the N-terminal fragment of ZNF185 (aa 1-200)." in p.8.

2. In Fig 2d (right panel), there are bands showing up in control lanes of the IP: Myc gel that are not present in the INPUT. I.e. the Myc antibody has positive bands in non-Myc-ZNF185 stimulated cell lanes, and the Flag

antibody similarly has positive bands in the non-PKA-Flag stimulated cell lanes. Can the authors clarify what are the exact bands the reader should focus on in the IP?

Response:

We thank the reviewer's important comments. To reduce background and non-specific bands, we used Myc-Tag (71D10) Rabbit mAb (Cell Signaling Technology) instead of Myc-Tag (9B11) Mouse mAb (Cell Signaling Technology), as shown in figure below. Additionally, we used goat anti-mouse light chain antibody (AP200A, Sigma-Aldrich) as the secondary antibody to delete IgG heavy-chain bands at 50 kDa.

A black arrow indicates light chain bands. Band intensities of Flag (PKA RII α -ZNF185 interaction) indicated by a red arrowhead were quantified.

3. In Fig 3d and e the authors examine the effect of ZNF185 loss on stress fibers and VE-cadherin in an endothelial monolayer. Can the authors explain what the take-home message is from this same experiment on single cells (Fig 3b)? To me the monolayer seems more informative in terms of permeability so perhaps the single cell data can be moved to supplement?

Response:

We thank you for your comments. As the reviewer pointed out, the single cell data in Fig. 3b was not informative compared with the endothelial monolayer data. Fig.3b was moved to Supplementary Fig.5.

4. Fig 3d and e. Fig 3d shows changes in VE-cadherin and F-actin expression in the monolayer. The resolution of this data needs to be improved. For example, the arrows indicate zig-zag junctions and the arrowheads gaps. This is difficult to see in these images and therefore I suggest adding high res, high-magnification data of these junctional phenotypes.

Fig 3 e shows the distribution of F-actin along cell-cell junctions. Although this is visually a really nice way to show that junctions are either zig-zag (irregular) or straight, this data is not quantitative as suggested in the Figure legend. To show whether there are changes in stress fiber distribution the authors really need to quantify F-actin expression (junctional vs non-junctional) across multiple junctions from multiple experiments. This will allow statistical testing of this phenotype between the different conditions.

I agree with the authors that there is a qualitative difference in stress fibers when comparing Sh-ZNF185 versus Sh-scramble. This is remarkably improved with forskolin treatment in the images, which does not seem as clearly reflected in panel e. Again, highlighting the need for more extensive quantitative analysis of this data.

A final suggestion related to this section of the story is for the authors to elaborate more on VE-cadherin

junctional integrity in the main text, which focusses almost exclusively on the actin, yet effects on VE-cadherin are prominent (at least qualitatively).

Response:

We appreciate the reviewer's comments. We captured the images in higher magnification and added the extended images to show the junctional phenotype, as shown in figures below.

In the main text, we also added “We further evaluated VE-cadherin dynamics at endothelial junctions. Immunofluorescence staining of VE-cadherin revealed that ZNF185 knockdown formed discontinuous zig-zag junctions indicated by white arrows with inter-endothelial gap junctions indicated by white arrowheads (Fig. 3b, Supplementary Fig. 6c).” in p.9.

Intensity of stress fibers at the yellow lines in above figures was quantified, as shown in figures below. The bar graph shows total amount of stress fiber intensity. As the reviewer pointed out, forskolin prevented stress fiber formation in HUVECs transduced with scrambled shRNA. By contrast, forskolin did not improve actin stress fiber formation in ZNF185 knockdown HUVECs.

We further have done quantitative analysis of the stress fiber intensity from multiple independent experiments, as shown in figures below. We added these data in Fig.3 and Supplementary figure 6.

5. In Fig 4d-e, what is the effect of RhoA inactivation (Y27632 treatment) on Sh-scramble control cells? This control should be included and quantified.

Response:

We thank you for your thoughtful comments. Y27632 has been reported to decrease endothelial permeability (PMID: 18097057, PMID: 28316141). Consistent with published reports, Y27632 inhibited baseline stress fiber formation and reduced endothelial permeability in sh-scrambled control cells, as shown in figures below.

Quantification of F-actin expression

We added "Inhibition of RhoA/Rho-associated protein kinase signaling by Y-27632 inhibited baseline stress fiber assembly (Fig. 4d, Supplementary Fig. 7a, b) and endothelial permeability (Fig. 4e) in sh-scrambled HUVECs, as previously reported^{27,28}." in p.11.

6. What is the recombinant ZNF185 referred to in Fig 5? It is unclear if this is referencing the C-terminal mutant version of ZNF185 shown in Supplement Fig 2 or a different recombinant version of the protein.

Response:

We appreciate your thoughtful comments. Immunoprecipitated Myc-full-length ZNF185-HA was used as the recombinant ZNF185. As shown in figure below, asterisks indicate full-length ZNF185. Thrombin divided full-length ZNF185 into Myc-N-terminal ZNF185 indicated by arrow and C-terminal ZNF185 indicated by arrowhead.

We added “Immunoprecipitated recombinant Myc-ZNF185-HA was also cleaved by thrombin (Fig. 7d).” in p.13.

We also added “Anti-HA beads were used to immunoprecipitate overexpressed Myc-ZNF185-HA in HUVECs. After immunoprecipitation, 10 U/ml thrombin was administered to recombinant ZNF185 for 1 h.” in figure legends.

7. Finally, in Fig 5 and associated text, the authors mention that cleavage of ZNF185 by thrombin may impair ZNF185 function similar to that of ZNF185 knockdown, causing RhoA activation and stress fibers. The authors could substantiate this hypothesis nicely by investigating whether expression of a ZNF185 variant that lacks the thrombin cleavage site in ZNF185 knockdown cells can rescue the effects of thrombin of EC junctions.

Response:

We appreciate your invaluable comments. Your suggestion helped us to greatly improve our manuscript. We generated stable cell lines of HUVECs overexpressing full-length ZNF185 or ZNF185-Δexon 11,12 (without cleavage site). Although thrombin induced stress fiber formation and the disruption of adherens junctions (indicated by white arrowheads) in HUVECs overexpressing full-length ZNF185, ZNF185-Δexon 11,12 rescued these disruptive effects of thrombin, as shown in figure below.

We added “To investigate the role of ZNF185 cleavage in HUVECs, we generated stable cell lines overexpressing full-length ZNF185 or ZNF185-Δexon 11,12. Although thrombin induced stress fiber formation and the disruption of endothelial adherens junctions in HUVECs overexpressing full-length ZNF185, ZNF185-Δexon 11,12 rescued these disruptive effects of thrombin (Fig. 8a, Supplementary Fig. 9a, b).” in p.13.

Minor comments:

1. In Fig 1d, immunofluorescence showed localisation of Myc-tagged ZNF185 with pPKA substrates to the membrane region upon forskolin treatment. These images are convincing, but the observations would be

strengthened by higher resolution images with a counterstain for VE-cadherin or a membrane marker.

Response:

We thank the reviewer's important comments. We obtained confocal images using microscope with a 63 × oil objective. To convert confocal images to PDF without losing quality, we used a TIFF image instead of a JPEG image. High resolution images with a counterstain for VE-cadherin in revised Fig.1e clearly show that phosphorylated Myc-tagged ZNF185 was localized at membrane region in response to forskolin treatment.

2. In Fig 2a, it might be helpful to indicate which pPKA substrates band is ZNF185. From the IP:HA gel it is clear that this is the ~90kDa band, but the multiple bands in the INPUT gel can be distracting without an arrowhead or asterisk at the bands of interest?

Response:

We appreciate the reviewer's comment on this point. We added black and red arrowheads which indicate bands of ZNF185. Band intensities of pZNF185 indicated by a red arrowhead were quantified.

3. In Fig 3a, the bands slightly larger than actin appear in the IP : HA gel and can be quite distracting from the fainter actin bands ~37kDa. As I understand it, the purpose of these results is to show that with ZNF185 overexpression and with forskolin treatment, actin is localised to ZNF185. So, what are these dark bands and could these be removed? It also would be helpful if the actin levels here were quantified similar to in Fig 1c or 2a to clearly demonstrate how this is changing with forskolin treatment.

Response:

We thank you for your thoughtful comments. To delete non-specific bands, we changed anti-actin antibody from mouse anti-β-actin (A2228; Sigma–Aldrich Corporation) to rabbit anti-β-actin (#4970; Cell Signaling Technology). We also added quantification of actin–ZNF185 interaction.

4. Information on the sh-ZNF185 knockdown and associated scrambled sequences is missing from the methods section.

Response:

We appreciate the reviewer’s comment. We added the method how to generate shRNA lentivirus in p.25.

REVIEWERS' COMMENTS:

Reviewer #1 (Remarks to the Author):

The authors have satisfactorily answered all the questions and concerns expressed in the first revision and the manuscript now fulfills the criteria to be published in Communication Biology

Reviewer #2 (Remarks to the Author):

The authors have done an excellent job addressing the comments raised by the reviewers. Nothing more to add.

Reviewer #3 (Remarks to the Author):

The revised manuscript by de Suzuki et al has really nicely addressed my comments. I very much appreciate the additional controls that have been added and new high resolution imaging data. I suggest that the study in its current form is acceptable for publication.